# PQGAN: Product-Quantised Image Representation for High-Quality Image Synthesis

**Denis Zavadski, Nikita P. Tatsch, Carsten Rother**
Heidelberg University, Zuse School ELIZA
{name.surname}.iwr.uni-heidelberg.de

## Abstract

Product quantisation (PQ) is a classical method for scalable vector encoding, yet it has seen limited usage for latent representations in high-fidelity image generation. In this work, we introduce *PQGAN*, a quantised image autoencoder that integrates PQ into the well-known vector quantisation (VQ) framework of VQ-GAN and adapts it to the regime of large-scale latent generative models. PQGAN achieves a noticeable improvement over state-of-the-art methods in terms of reconstruction performance, including both quantisation methods and their continuous counterparts. We achieve a PSNR score of 37 dB, where prior work achieves 27 dB, and are able to reduce the FID, LPIPS, and CMMD score by up to 96%. Our key to success is a thorough analysis of the interaction between codebook size, embedding dimensionality, and subspace factorisation, with vector and scalar quantisation as special cases. We obtain novel findings, such that the performance of VQ and PQ behaves in opposite ways when scaling the embedding dimension. Furthermore, our analysis shows performance trends for PQ that help guide optimal hyperparameter selection. Finally, we demonstrate that PQGAN can be seamlessly integrated into pre-trained diffusion models. This enables either a significantly faster and more compute-efficient generation, or a doubling of the output resolution at no additional cost, positioning PQ as a strong extension for discrete latent representations in image synthesis.

## 1 Introduction

Recent advancements in image processing in general and image generation in particular, have intensified the need for scalable solutions. Datasets grow larger (Schuhmann et al., 2022; Gadre et al., 2023), models become increasingly compute-intensive (Podell et al., 2024; Rombach et al., 2022) and resolutions increase (Chen et al., 2024b;a). Addressing this trend typically requires either a proportional increase in computational resources, which is often impractical, or more efficient data representations that reduce redundancy while preserving semantic content (Rombach et al., 2022; Peebles & Xie, 2023; Esser et al., 2021). A common strategy is to operate in the latent space of an autoencoder trained for image reconstruction, under the assumption that this bottleneck representation discards perceptually negligible details while retaining the core visual structure (Rombach et al., 2022; Lee et al., 2022).

VQ-VAE (Van Den Oord et al., 2017) demonstrated that latent image representations can be quantised by learning a codebook of discrete vectors, where each latent vector is replaced by its nearest codebook entry. This mapping transforms the latent space into a discrete index space, where each spatial location is represented by a single codebook index. Crucially, this makes the storage cost of the representation independent of the dimensionality of the vectors themselves and only the spatial resolution and the codebook size determine the representation size. This property enabled a new class of models for image compression (El-Nouby et al., 2023; Mentzer et al., 2020; 2024) and autoregressive generation (Esser et al., 2021; Ramesh et al., 2021; Yu et al., 2022b), where discrete latents offer both compactness and modelling flexibility.

While it is theoretically possible to use arbitrarily high-dimensional latent vectors, learning such codebook entries becomes increasingly difficult as dimensionality grows (Zhu et al., 2024). Because for VQ all dimensions are coupled within each codebook entry, the training signal becomes sparse

| Input | PQGAN (F = 8) | PQGAN (F = 16) | SDv2.1 VAE (F = 8) |

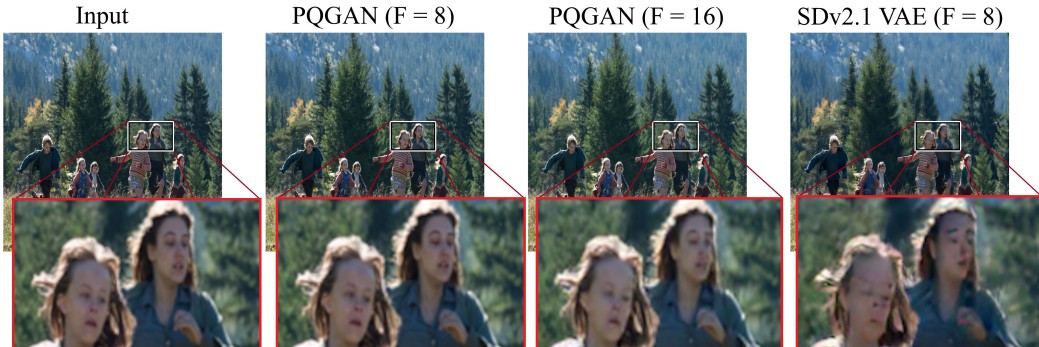

Figure 1: Reconstruction results for our quantised PQGAN vs. the continuous VAE in StableDiffusion2.1 (Rombach et al., 2022). Even with stronger spatial downsizing factor $F$ for the latent representation, our product quantised representation space shows higher fidelity (see details in faces).

in high-dimensional spaces, leading to slower convergence and limiting the effective size of the codebook. Moreover, redundancy often arises, as distinct entries may differ only partially across dimensions. We address both issues by applying product quantisation (PQ), originally introduced by Jegou et al. (2010), and factorising each codebook entry into $S$ independent subspaces. Learning each subspace independently reduces the dimensionality of the learned features and ensures that training signals remain dense and well-conditioned. The factorisation enables our model to represent a fictive codebook of size $K^S$, where $K$ is the number of entries per subspace. Since entries from different subspaces can be composed freely, there is no need to store or learn redundant combinations explicitly. These properties allow us to achieve state-of-the-art results on ImageNet (Russakovsky et al., 2015), with FID scores as low as 0.036 and PSNR values up to 37.4 dB, using only 512 codebook entries per subspace.

In this work, we conduct a thorough evaluation of PQ for latent image representation across its full hyperparameter spectrum, spanning the extremes of vector and scalar quantisation. Our analysis reveals consistent trends that guide principled parameter selection. Furthermore, it gives novel findings, such that the performance of VQ and PQ behaves in opposite ways when scaling the embedding dimension.

In applications such as image compression (Mentzer et al., 2020; El-Nouby et al., 2023) and autoregressive modelling (Esser et al., 2021), PQ introduces practical challenges: As the number of subspaces $S$ increases, one must either store $S$ separate codebook indices or construct a joint codebook, which becomes exponentially large and unfeasible to store for compression or learn for generative models. As a result, prior work has not fully explored the representational capacity of PQ. However, we observe that the limitation of having a small set of codebook indices does not apply in the context of diffusion- and flow-based generation (Rombach et al., 2022; Podell et al., 2024; Ho et al., 2020; Esser et al., 2024), where the decoder operates directly in the latent space without requiring an explicit autoregressive factorisation. This makes PQ particularly well-suited for diffusion- and flow-based models, and allows us to explore its full potential in a large-scale generative setting.

We demonstrate that large latent diffusion models, such as Stable Diffusion (Rombach et al., 2022), can operate effectively on spatially restricted yet high-dimensional latent representations, maintaining comparable or even reduced computational cost. However, our evaluation reveals that as image resolution increases, the reconstruction quality of default latent representations can degrade significantly, introducing artefacts that propagate through the generative process while PQ representations manage to maintain a high reconstruction fidelity (compare Figure 1).

Our contributions can be summarised as follows:

- We propose PQGAN, a new latent representation for image generation, based on the concept of product quantisation (PQ).

- We conduct a thorough analysis of PQ hyperparameters, with Vector and Scalar quantisation as special cases. We observe novel findings, such that the performance of VQ and PQ

behaves in opposite ways when scaling the embedding dimension and further performance trends that guide optimal hyperparameter selection.

- We demonstrate that PQ enables state-of-the-art image representation quality, achieving a PSNR of 37.4 dB and FID as low as 0.036 using only 512 codebook entries per subspace, surpassing even continuous counterparts.

- We show that pre-trained diffusion models can be efficiently adapted to PQ-based representations, enabling either significantly faster generation or double resolution at equal cost.

## 2 RELATED WORK

**Latent Image Representation for Generative Models.** Modern generative models demand increasingly high computational resources, driven by the scaling of model parameters (Podell et al., 2024; Peebles & Xie, 2023), the use of attention-heavy (Vaswani et al., 2017) architectures (Chen et al., 2024b; Peebles & Xie, 2023; Touvron et al., 2021), and the pursuit of high-resolution image synthesis (Podell et al., 2024; Chen et al., 2024a). Many of the most successful methods, such as autoregressive models and diffusion models, also require iterative inference, further compounding the cost of generation. Optimising the generative pipeline remains an active area of research, either by improving the generative process itself (Sauer et al., 2024b; Kang et al., 2024; Salimans & Ho, 2022; Lin et al., 2024; Sauer et al., 2024a) or by modifying the representation on which the model operates. The latter strategy typically involves encoding high-resolution images into lower-resolution latent spaces, significantly reducing the computational burden during generation. Different generative models impose different constraints on their latent spaces. Autoregressive transformers, for example, require discrete and indexable representations (Esser et al., 2021), hereby limiting the capacity of the underlying representation. However, diffusion models are generally assumed to operate effectively on continuous latent spaces (Rombach et al., 2022; Podell et al., 2024; Pernias et al., 2024; Chen et al., 2024a), provided they are sufficiently low-dimensional. In practice, this often motivates the use of both low spatial resolutions and reduced channel dimensionality in the latent representation to ensure tractable generation. Contrary to common practice (Rombach et al., 2022; Podell et al., 2024), we find that diffusion models can efficiently operate on high-dimensional embeddings as long as the spatial resolution remains low. By leveraging PQ, we learn a discrete representation that not only surpasses continuous latent spaces in reconstruction fidelity, but also allows for even lower spatial resolutions - resulting in more efficient generation with no loss in quality.

**Vector Quantisation and Codebook Models.** Vector quantisation (VQ), popularised by VQ-VAE (Van Den Oord et al., 2017) learns discrete latent representations by replacing each continuous vector with its nearest entry from a learned codebook. This approach is well-suited to autoregressive models (Esser et al., 2021; Lee et al., 2022; Yu et al., 2022b), and has also been widely applied in image compression (Mentzer et al., 2020; 2024), where discrete indices compactly represent high-dimensional latent vectors.

However, standard VQ suffers from limitations when applied to high-dimensional embeddings. Because all dimensions are quantised jointly through a single codebook lookup, the model receives sparse and entangled training signals, often leading to codebook collapse and slow convergence. To address this, various extensions have been proposed (Yu et al., 2022a) - such as residual quantisation (Kim et al., 2025; Lee et al., 2022), hierarchical quantisation (Razavi et al., 2019), and other regularisation techniques (Zhang et al., 2023) - but these generally offer only incremental improvements and retain the disadvantage of high dimensional codebooks with sparse training signals.

One core issue remains: the joint modelling of high-dimensional latent spaces via a single codebook imposes a structural bottleneck. Multi-channel quantisation (Zheng et al., 2022) made the first step into splitting the latent representation along the channel side and used one joined codebook for all splits thereby limiting expressiveness and the number of reasonable splits. In the context of image compression, scalar quantisation and product quantisation have been explored (El-Nouby et al., 2023; Mentzer et al., 2020), but their capacity was limited by the need to balance bitrate and distortion.

In contrast, we apply PQ in the context of latent representation learning for diffusion-based generative models. Freed from rate-distortion constraints, we explore the full potential of PQ and show that a highly factorised latent space not only avoids the training difficulties of conventional VQ,

but also enables reconstructions that surpass even continuous autoencoders (Rombach et al., 2022; Podell et al., 2024) - achieving state-of-the-art fidelity while operating at significantly lower spatial resolution.

## 3 METHOD

In this work, we isolate the effect of product quantisation on the representational capacity of latent autoencoders. To ensure a controlled comparison, we adopt the VQGAN (Esser et al., 2021) architecture and training pipeline, widely regarded as the standard backbone for vector-quantised image representation, without modification. Our only change is to replace the original vector quantisation module with a product quantisation scheme, as detailed in Section 3.1, enabling a direct evaluation of PQ's impact on reconstruction quality.

To further assess the utility of these representations, we train StableDiffusion2.1 (Rombach et al., 2022) to operate directly on our product quantised latent space and revisit two common assumptions in diffusion-based image generation. First, we challenge the widespread misconception that latent representations must be kept low-dimensional to maintain computational efficiency. Second, we highlight a critical oversight: as image resolution increases, the reconstruction quality of standard latent spaces deteriorates, significantly introducing artefacts that propagate through the generative process. We discuss both issues in detail in Section 3.2.

### 3.1 PRODUCT QUANTISATION

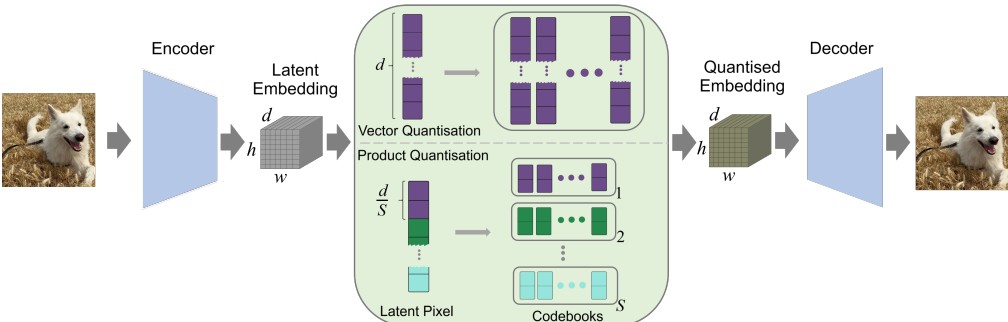

Figure 2: Comparison of vector quantisation (VQ, top) and product quantisation (PQ, bottom) in the latent space of an autoencoder. VQ replaces each latent vector with the nearest entry from a single codebook of size $K$. PQ splits the vector into $S$ subspaces, quantising each independently, resulting in a combinatorially large fictive codebook of size $K^S$ while maintaining low per-subspace complexity.

**Preliminaries – Vector Quantisation.** We begin by formalising the notation for vector quantisation in the context of latent image representations. Let $x \in \mathbb{R}^{H \times W \times 3}$ denote an RGB image of resolution $H \times W$. Its latent embedding $z_e$ is obtained from the bottleneck of an autoencoder, typically with a spatial resolution reduced by a downsampling factor $F$. The resulting latent representation is $z_e \in \mathbb{R}^{h \times w \times d}$, where $h = H/F$, $w = W/F$, and $d$ is the embedding dimensionality. We refer to each vector at a spatial location as a latent "pixel", denoted by $p_\ell \in \mathbb{R}^d$.

To discretise this representation, vector quantisation learns a codebook $C = \{e_1, \ldots, e_K\}$ consisting of $K$ codewords $e_k \in \mathbb{R}^d$. Each latent vector $p_\ell$ is then replaced by its nearest codebook entry under some distance metric (typically $\ell_2$). However, learning such a high-dimensional codebook is non-trivial: since all $d$ dimensions must be learned jointly, the signal-to-parameter ratio becomes increasingly sparse, which limits scalability and leads to slower convergence (see Section 4.2).

**Product Quantisation.** To address the challenges of high-dimensional codebook learning, PQ decomposes each latent vector $p_\ell \in \mathbb{R}^d$ into $S$ disjoint subspaces of dimension $d/S$ (assuming $d$ is divisible by $S$), and learns a separate codebook for each subspace. That is, for $S \in \{1, \ldots, d\}$, each latent pixel is split as $p_\ell = [p_\ell^{(1)}, \ldots, p_\ell^{(S)}]$ with $p_\ell^{(s)} \in \mathbb{R}^{d/S}$, and each $p_\ell^{(s)}$ is quantised indepen-

dently using its own codebook $C^{(s)} = \{e_1^{(s)}, \ldots, e_K^{(s)}\}$. Vector and scalar quantisation emerge as special cases of PQ when $S = 1$ and $S = d$, respectively.

A visual comparison between VQ and PQ is shown in Figure 2. Because the sub-codebooks are trained independently, the signal per parameter is significantly denser, improving convergence and scalability. Moreover, since sub-codebook entries can be freely recombined across subspaces, PQ defines a *fictive* codebook of combinatorial size $K^S$ without requiring the storage or learning of all combinations individually. This greatly increases the expressive capacity of the representation while keeping each sub-codebook manageable in size.

More formally, the advantages of PQ can be attributed to two closely related phenomena:

1. **Training signal sparsity and sample efficiency.** Vector quantisation operates on the full entangled $d$-dimensional latent space, and learning a meaningful codebook requires many training samples to sufficiently populate that space. To achieve a covering resolution $\epsilon$, the required number of samples grows as $\mathcal{O}((1/\epsilon)^d)$, leading to severe data sparsity in high dimensions. This impairs codebook learning, as each centroid receives little supervision. In contrast, PQGAN factorises the space into $S = d/2$ subspaces of dimension 2, each of which can be learned independently with only $\mathcal{O}((1/\epsilon)^2)$ samples to achieve the same latent coverage resolution of $\epsilon$, regardless of the latent dimension. This keeps the effective sample density constant as $d$ increases, allowing PQ to scale latent dimensionality without sacrificing codebook fidelity.

2. **Quantisation error scaling.** Classical quantisation theory (Zador, 1982) shows that mean squared quantisation error scales as $\mathcal{O}(K^{-2/d})$ in $d$-dimensional space with $K$ centroids. For VQ, this implies that increasing latent dimension requires exponential growth in codebook size to maintain the same quantisation error, which is an impractical trade-off. PQ sidesteps this by operating in fixed low-dimensional subspaces (with $S = d/2$), where the quantisation error remains constant $\mathcal{O}(K^{-2/2}) = \mathcal{O}(K^{-1})$, independent of total latent dimension. This allows PQ to increase total representational capacity (via higher $d$) without degrading quantisation error or requiring larger codebooks.

Together, these principles suggest a clear difference in scaling behaviour. While VQ is expected to degrade as latent dimensionality increases, PQ should benefit from higher-dimensional embeddings. These trends arise naturally from the differing scaling properties of VQ and PQ with respect to training signal efficiency and quantisation error.

As in standard VQ, the quantisation objective minimises the reconstruction loss between the original and quantised latents, and includes a commitment term to stabilise codebook usage. The training objective is:

$$\mathcal{L}_{\text{PQ}} = \|z_e - \text{sg}(z_q)\|_2^2 + \beta\|\text{sg}(z_e) - z_q\|_2^2 + \mathcal{L}_{\text{rec}} + \lambda_{\text{adv}}\mathcal{L}_{\text{GAN}}, \tag{1}$$

where $\text{sg}(\cdot)$ denotes the stop-gradient operator, $z_q$ is the quantised latent, $\mathcal{L}_{\text{rec}}$ is the perceptual reconstruction loss, and $\mathcal{L}_{\text{GAN}}$ is the adversarial loss used in VQGAN training. All components are inherited unchanged from the original VQGAN (Esser et al., 2021) framework.

## 3.2 LATENT ADAPTATION

Modern state-of-the-art image generation models typically operate on latent representations rather than raw images to improve computational efficiency (Rombach et al., 2022; Podell et al., 2024). The key factor behind this efficiency is the reduced *spatial* resolution of the latent space. In diffusion models such as Stable Diffusion (Rombach et al., 2022), the use of attention layers throughout the architecture causes computational cost to scale quadratically with spatial resolution. As a result, generative models usually favour low-dimensional latents to limit compute.

However, we highlight that in diffusion models, *spatial resolution* - not channel dimensionality - is the primary computational bottleneck. We argue that high-dimensional latent representations can be used without compromising efficiency. To validate this, we adapt StableDiffusion2.1 to operate directly on product-quantised latents in a two-stage training procedure.

We observe that the U-Net generator contains a fixed projection from 4 to 512 channels at the input layer, and from 512 back to 4 channels at the output layer. These projections form an artificial bottleneck that restricts the usable dimensionality of the latent space. We increase these projections to match the dimensionality of our quantised latents, leaving the rest of the architecture unchanged. In the first stage, we train only the modified input and output projections, keeping all other weights frozen. After 20k steps, we unfreeze the entire model and continue fine-tuning for an additional 1M steps with the typical diffusion objective

$$\mathcal{L} = \mathbb{E}_{z_0, t, c_t, \epsilon \sim \mathcal{N}(0,1)} \left[ \| \epsilon - \epsilon_\theta(z_t, t, c_t) \|_2^2 \right], \tag{2}$$

with the target image $z_0$, the noisy image $z_t$, the timestep $t$, the text conditioning $c_t$ and noise $\epsilon$.

## 4 EXPERIMENTS

The first part of our experimental analysis investigates how the key design choices in product quantisation - namely, codebook size, subspace factorisation, and latent dimensionality - influence reconstruction quality and codebook behaviour. In Section 4.2, we present a structured evaluation that spans the quantisation spectrum between vector and scalar regimes, revealing consistent trends and points of saturation in model performance.

To assess the efficiency and stability of the learned representations, we further analyse codebook utilisation via entropy and perplexity metrics in Section 4.3. Finally, in Section 4.4, we compare our models to state-of-the-art discrete and continuous latent representations, demonstrating clear and consistent gains in both reconstruction fidelity and quantisation robustness.

In the second part of our evaluation (Section 4.5), we adapt StableDiffusion2.1 to operate on our product-quantised latent space. This setup allows us to examine two underexplored assumptions in latent diffusion models: the need for low-dimensional latents and the presumed absence of degradation of reconstruction quality at high resolution. We show that product-quantised latents offer practical advantages in both computational efficiency and fidelity, particularly in the high-resolution regime.

### 4.1 METRICS

To evaluate the quality of the reconstructed images, we report four complementary metrics: FID (Heusel et al., 2017), PSNR, CMMD (Jayasumana et al., 2024), and LPIPS (Zhang et al., 2018). While FID remains a widely adopted perceptual metric, it has known limitations due to its reliance on Gaussian assumptions and sensitivity to distributional shifts (Jayasumana et al., 2024). To address this, we additionally employ CMMD, which has been shown to provide a more faithful and unbiased comparison of image similarity.

For pixel-level accuracy and structural fidelity, we report PSNR and LPIPS, which respectively quantify exact reconstructions and perceptual consistency. Finally, to assess the effectiveness of our product-quantised representations, we analyse codebook utilisation using two complementary metrics - normalised entropy $H_n$, which captures the uniformity of code usage, and normalised perplexity $P_n$, which reflects the fraction of entries actively used:

$$H_n = \frac{H(p)}{\log K} = \frac{-\sum_i^K p_i \log p_i}{\log K}, \qquad P_n = \frac{P(p)}{K} = \frac{\exp(H(p))}{K}, \tag{3}$$

where $p$ is the empirical codeword distribution and $H(p)$ is its Shannon entropy. Both metrics are bounded in $[0, 1]$, with $H_n = 1$ indicating perfectly uniform usage over the active subset of the codebook, and $P_n = 1$ denoting full codebook activation.

### 4.2 PRODUCT SPACE COMPOSITION

To study the interaction between key quantisation parameters, we systematically vary the latent dimensionality $d \in \{4, \ldots, 256\}$, the number of subspaces $S \in \{1, \frac{d}{2}, \frac{d}{4}, \frac{d}{8}, d\}$, and the codebook

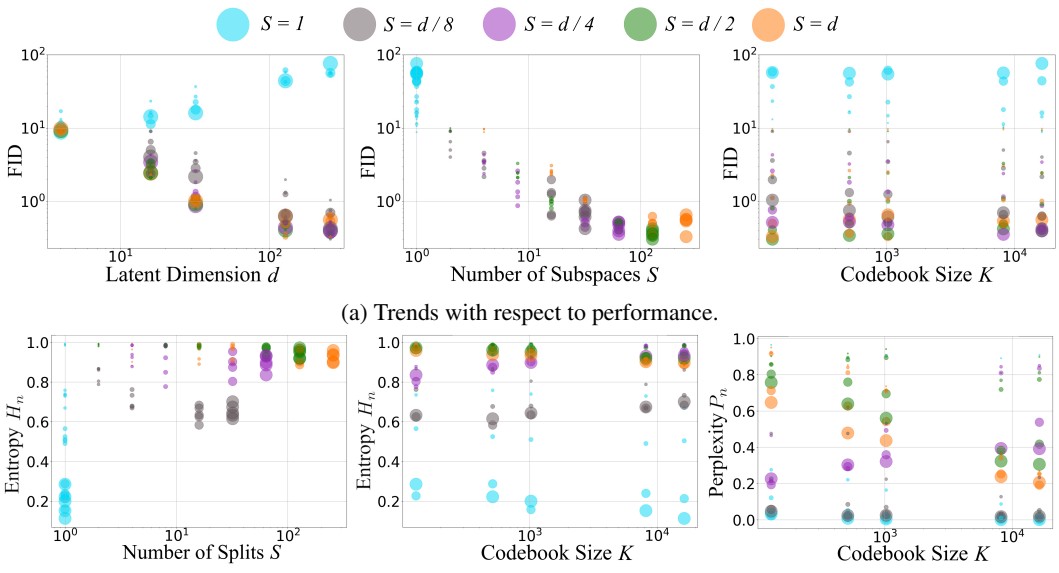

(a) Trends with respect to performance.

(b) Trends with respect to codebook utilisation.

Figure 3: Quantitative analysis of product quantisation configurations. (a) Performance across latent dimensionality $d$, number of subspaces $S$, and codebook size $K$ as measured by FID. (b) Codebook utilisation metrics (entropy and perplexity) for the same configurations, showing improved usage and efficiency with increasing factorisation.

size $K \in \{128, 512, 1024, 8192, 16384\}$. All models are trained on the ImageNet (Russakovsky et al., 2015) dataset at a resolution of 256×256 for one million steps using a batch size of 20.

Figure 3a presents FID scores on a log–log scale across different PQ configurations. Each marker represents a configuration, with colour indicating the number of subspaces $S$. In the left plot, marker size reflects codebook size $K$; in the middle and right plots, it reflects latent dimensionality $d$.

We observe a clear divergence in behaviour between vector quantisation (Esser et al., 2021) ($S = 1$) and PQ: increasing latent dimensionality degrades performance for VQ, while improving it for PQ (left). Increasing the number of subspaces $S$ leads to consistent performance gains, saturating around $S = d/2$, just before the scalar quantisation limit at $S = d$ (middle).

FID appears only weakly dependent on codebook size, but is strongly influenced by the number of subspaces (right). The plot shows that even for small codebooks ($K = 128$ or 512), PQ models with higher dimensionality and greater factorisation outperform all VQ configurations.

These results indicate that transitioning from VQ to PQ fundamentally changes the scaling behaviour of latent representations: PQ benefits from larger latent dimensions and learns high fidelity, whereas VQ fails to scale effectively. Full results for PSNR, LPIPS, and CMMD are provided in the supplement C and confirm the same trends. However, maintaining a high number of subspaces requires higher codebook matching times that scale linearly with the number of subspaces which can put a practical limitation on the usability of PQ in high dimensional settings. The best performing PQ autoencoder in our experiments had a quantisation overhead of just 50% of walltime compared to when codebook matching is omitted. More details on the computational overhead are provided in the supplement B.

## 4.3 CODEBOOK UTILISATION

One of the main challenges in vector quantisation is learning high-dimensional latent representations within a large codebook. Therefore we track codebook diagnostics to assess the effective utilisation and diversity of the quantised representations. Figure 3b shows the normalised entropy $H_n$ and the normalised perplexity $P_n$ for all evaluated configurations, plotted on a linear–log scale. Marker colours indicate the number of subspaces $S$, and marker sizes correspond to latent dimensionality $d$.

Table 1: Comparison of quantisation methods on ImageNet 256×256 validation set. Metrics include reconstruction FID (rFID), PSNR, CMMD, LPIPS and information about the downsampling factor **F**, the channel latent dimension $d$, codebook size **K** and the latent training method with KL, vector quantisation (VQ), residual quantisation (RQ) product quantisation (PQ) and multi-channel quantisation (MCQ).

| Method | Latent | F | Latent Resolution | $d$ | K | PSNR↑ | rFID↓ | CMMD↓ | LPIPS↓ |
|---|---|---|---|---|---|---|---|---|---|
| VQGAN (Esser et al., 2021) | VQ | 16 | $16 \times 16$ | 256 | 16 384 | 19.7 | 4.98 | 0.422 | 0.1633 |
| | VQ | 16 | $16 \times 16$ | 256 | 1 024 | 19.4 | 7.94 | 0.862 | 0.1834 |
| VQGAN (LDM (Rombach et al., 2022)) | VQ | 16 | $16 \times 16$ | 8 | 16 384 | 20.6 | 5.51 | 0.621 | 0.1462 |
| | VQ | 8 | $32 \times 32$ | 4 | 16 384 | 23.1 | 1.29 | 0.258 | 0.0815 |
| | VQ | 4 | $64 \times 64$ | 3 | 256 | 26.4 | 0.47 | - | - |
| VQGAN-LC (Zhu et al., 2024) | VQ | 16 | $16 \times 16$ | 8 | 100 000 | 23.8 | 2.62 | 0.240 | 0.1280 |
| | VQ | 8 | $32 \times 32$ | 4 | 100 000 | 27.0 | 1.29 | 0.080 | 0.0712 |
| ViT-VQGAN (Yu et al., 2022a) | VQ | 8 | $32 \times 32$ | 32 | 8 192 | - | 1.28 | - | - |
| Mo-VQGAN (Zheng et al., 2022) | MCQ | 16 | $16 \times 16$ | 256 | 1 024 | 22.4 | 1.12 | - | 0.1132 |
| RQ-VAE (Lee et al., 2022) | RQ | 32 | $8 \times 8$ | 256 | 16 384 | - | 2.69 | - | - |
| LDM VAE (Rombach et al., 2022) | KL | 16 | $16 \times 16$ | 16 | - | 24.0 | 0.87 | 0.198 | 0.0665 |
| | KL | 8 | $32 \times 32$ | 4 | - | 24.2 | 0.94 | 0.168 | 0.0651 |
| | KL | 4 | $64 \times 64$ | 3 | - | 28.1 | 0.27 | 0.080 | 0.0282 |
| SDv2.1 VAE (Rombach et al., 2022) | KL | 8 | $32 \times 32$ | 4 | - | 25.3 | 0.75 | 0.133 | 0.0610 |
| SDXL VAE (Podell et al., 2024) | KL | 8 | $32 \times 32$ | 4 | - | 25.3 | 0.74 | 0.148 | 0.0573 |
| PQGAN (ours) | PQ | 32 | $8 \times 8$ | 256 | 256 | 24.6 | 0.90 | 0.164 | 0.0621 |
| | PQ | 16 | $16 \times 16$ | 128 | 128 | 28.3 | 0.41 | 0.094 | 0.0304 |
| | PQ | 8 | $32 \times 32$ | 128 | 512 | **37.4** | **0.036** | **0.011** | **0.0024** |

VQ ($S = 1$) exhibits the lowest entropy and fails to scale with increasing latent dimensionality, as reflected in declining performance for larger $d$ (larger marker sizes, left).

In contrast, increasing the number of subspaces $S$ leads to more uniform code usage, with entropy $H_n$ largely unaffected by codebook size $K$ (middle). Instead, we observe that entropy clusters by subspace configuration, reflecting the structure imposed by PQ. The right plot shows that normalised perplexity $P_n$, unlike entropy, does depend on $K$: as $K$ increases, high-dimensional PQ representations gradually saturate in their code usage. Even for very large codebooks ($K = 16\,384$), PQ models maintain $H_n > 0.8$, suggesting robust and effective utilisation despite overparameterisation.

These results reinforce our earlier findings: PQ not only improves reconstruction performance with increasing factorisation, but also makes more efficient use of the available codebook capacity, while VQ struggles to scale beyond small latent dimensions.

## 4.4 COMPARISON TO STATE OF THE ART

Building on our previous analysis, we select the best-performing configuration at downsampling factor $F = 16$, which uses $d = 128$ channels, $S = 64$ subspaces, and a codebook size of $K = 128$. For $F = 8$, we evaluate the top three configurations and retain the best ($S = 64$, $d = 128$, $K = 512$) as our final high-fidelity $F = 8$ model. To allow for a fair comparison to state-of-the-art residual quantisation (Lee et al., 2022), we also train a $F = 32$ model with the same dimension $d = 256$ as the competitor but with $S = 64$ and $K = 256$. Table 1 compares reconstruction performance on the ImageNet (Russakovsky et al., 2015) validation set against state-of-the-art quantisation methods as well as continuous latent spaces based on KL-regularised autoencoders within the VQGAN framework. We also include the latent representations employed by the production-grade diffusion models SDv2.1 (Rombach et al., 2022) and SDXL (Podell et al., 2024).

For the ImageNet $256 \times 256$ comparison, we evaluate latent spaces downsized by factors $F \in \{8, 16, 32\}$, resulting in latent resolutions of $32 \times 32$, $16 \times 16$ and $8 \times 8$, respectively. We also report results for $F = 4$ (i.e., $64 \times 64$ latent resolution), but highlight these in grey to indicate that such configurations are impractical for real-world applications and primarily serve as performance upper bounds for competing methods.

Our best-performing model, PQGAN with $F = 8$, achieves a PSNR of 37.4 dB - substantially outperforming all other quantised and continuous baselines across all evaluation metrics. According to rFID, CMMD, and LPIPS, reconstructions from this model are nearly indistinguishable from the original images. Notably, the second-best model is also PQGAN, operating at $F = 16$, i.e. with half

Table 2: Comparison of representation transferability for different encoding methods.

| Method | Latent Resolution | FFHQ | | | | LSUN | | | |
|---|---|---|---|---|---|---|---|---|---|
| | | rFID↓ | PSNR↑ | CMMD↓ | LPIPS↓ | rFID↓ | PSNR↑ | CMMD↓ | LPIPS↓ |
| VQGAN-LC (Zhu et al., 2024) | $32 \times 32$ | 3.35 | 29.6 | 0.094 | 0.0390 | 7.01 | 24.7 | 0.261 | 0.0711 |
| SDv2.1 VAE (Rombach et al., 2022) | $32 \times 32$ | 1.57 | 29.9 | 0.086 | 0.0284 | 5.51 | 25.0 | 0.321 | 0.0629 |
| SDXL VAE (Podell et al., 2024) | $32 \times 32$ | 1.77 | 30.0 | 0.122 | 0.0284 | 6.00 | 24.9 | 0.317 | 0.0566 |
| PQGAN (Ours) | $16 \times 16$ | 0.54 | 33.3 | 0.190 | 0.0138 | 4.77 | 27.9 | 0.353 | 0.0315 |
| | $32 \times 32$ | **0.13** | **42.1** | **0.021** | **0.0012** | **0.62** | **38.3** | **0.189** | **0.0022** |

Table 3: Quantitative comparison of StableDiffusion (SD) and SD when adapted to PQ latent space. Sample time is measured on a A100 GPU with 50 sampling steps. Sample time and required memory is measured for the generation of one sample.

| Generator | Image Size | Latent Resolution | F | $d$ | Samples / s ↑ | Memory (GB) ↓ |
|---|---|---|---|---|---|---|
| SDv2.1 (Rombach et al., 2022) | $768 \times 768$ | $96 \times 96$ | 8 | 4 | 0.116 | 14.9 |
| PQSD-HR (Ours) | $1536 \times 1536$ | $96 \times 96$ | 16 | 128 | 0.112 | 14.9 |
| PQSD-Precise (Ours) | $768 \times 768$ | $96 \times 96$ | 8 | 128 | 0.116 | 14.8 |
| PQSD-Quick (Ours) | $768 \times 768$ | $48 \times 48$ | 16 | 128 | 0.465 | 7.7 |

the spatial resolution, yet still outperforming all competing methods, including those with $F = 4$, which use 16 times more latent pixels.

While other quantised approaches require large codebooks (up to $100\,000$ entries (Zhu et al., 2024)) to achieve competitive results, PQGAN attains superior performance with compact codebooks of $K = 128$ to $512$, depending on the downsampling factor. These results underscore the efficiency and scalability of product quantisation in high-fidelity latent representation learning.

To assess the generality of our learned representations, we evaluate their transferability to unseen datasets. Specifically, we compare PQGAN against the best-performing quantised and continuous baseline models (at $F = 8$) on FFHQ (Karras et al., 2019) (10k images) and LSUN (Yu et al., 2015) (3k images) resized to $256 \times 256$. Results are shown in Table 2.

PQGAN again achieves the strongest performance across all metrics, even when operating with a smaller latent resolution. On FFHQ, PSNR increases for all models, with PQGAN ($F = 8$) reaching 42.1 dB, while LPIPS also improves substantially. Although rFID and CMMD scores increase across the board, PQGAN remains top-ranked. Since both rFID and CMMD are global statistical metrics rather than image-pair comparisons, they are best interpreted for relative model ranking rather than absolute cross-domain comparison.

On LSUN, LPIPS values remain largely consistent with ImageNet results. The continuous baselines show an almost unchanged PSNR, while VQGAN-LC (Zhu et al., 2024) experiences a slight drop. PQGAN, by contrast, maintains or even slightly improves its performance, demonstrating robustness under domain shift across all metrics, including FID and CMMD.

These results suggest that PQGAN learns stable and expressive representations that generalise well to unseen data. Even when operating at lower spatial resolutions, it consistently outperforms both quantised and continuous baselines - highlighting the effectiveness of PQ for robust representations.

By training a latent diffusion model from scratch on the PQGAN representation with the ImageNet data, the models achieve an improved FID score for generated ImageNet samples compared to the model trained on the StableDiffusion2.1 VAE. More details are provided in the supplement D.

## 4.5 LATENT SPACE INTEGRATION

Having established the fidelity and generality of the learned representation, we now investigate its usability as a generative prior by integrating it into a StableDiffusion2.1 (Rombach et al., 2022). As discussed in Section 3.2, the default SD generator operates on a continuous latent space with downsampling factor $F = 8$ and dimensionality $d = 4$, mapping images from $(H \times W \times 3)$ to $(H/8 \times W/8 \times 4)$. In our setup, we explore three PQGAN-based variants summarised in Table 3, each offering different trade-offs between spatial resolution and generative efficiency.

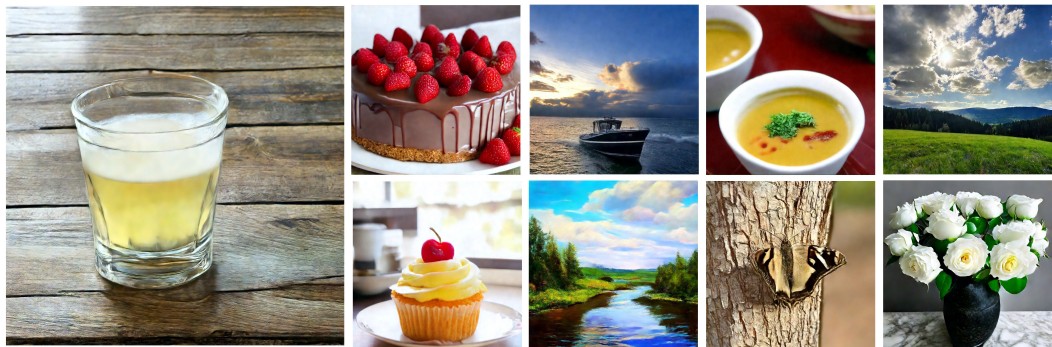

Figure 4: Generated examples for our PQSD-HR (left), PQSD-Precise (top), PQSD-Quick (bottom).

Because our PQGAN models were originally trained at lower resolutions, we fine-tune both $F = 8$ and $F = 16$ variants on ImageNet (Russakovsky et al., 2015) at $768 \times 768$ for 200,000 steps using a batch size of 15. Figure 1 shows a qualitative comparison against the native SD latent representation. The SD-VAE exhibits degradation at high resolution, including hallucinations, which are absent from our PQ-based reconstructions. More examples in the supplement E.

To adapt SD's U-Net architecture to higher channel dimensionality, we modify only the first and last convolutional layers, increasing their channel width while reusing the pretrained weights. To preserve the pretrained model's capabilities, we first freeze all other layers and train only the newly added components first for 20k steps. We then train the full model and gradually increase the resolution up to $768 \times 768$ for PQSD-Quick and $1536 \times 1536$ for PQSD-HR. Full training details are provided in the supplement A.

Figure 4 shows generations from all three PQSD variants. Generating $1536 \times 1536$ samples with PQSD-HR ($F = 16$) requires approximately the same computational cost as $768 \times 768$ generation with SDv2.1, demonstrating the efficiency of operating at lower spatial resolutions (see Table 3). Alternatively, PQSD-Quick allows $768 \times 768$ samples to be generated with a speed-up of up to 4 times, while maintaining higher reconstruction fidelity than the native SD-VAE. These results highlight that product-quantised latent spaces not only generalise well and scale efficiently, but can also serve as drop-in replacements in generative pipelines, offering improved performance with minimal architectural modification.

## 5 Conclusion & Discussion

We introduced PQGAN, a product-quantised image representation model that achieves state-of-the-art latent reconstruction fidelity. By integrating product quantisation into the VQGAN framework, we obtain high-dimensional latent spaces at reduced spatial resolution without sacrificing codebook efficiency.

We demonstrated that such representations can be seamlessly integrated into pre-trained diffusion models. Adapting Stable Diffusion to three distinct PQ configurations, we showed that large generative models can operate on new latent spaces without full retraining. This enables either a doubling of output resolution at comparable computational cost or faster generation at fixed resolution.

While product-quantised spaces are not directly suited for index-based tasks due to the exponential growth of index combinations with increasing splits, generative methods that do not rely on indexing but use quantisation (Kolesnikov et al., 2022) or image representations directly (Chen et al., 2024a; BFL et al., 2025) can fully benefit from PQ representations.

Overall, our results establish product quantisation as a practical and scalable alternative to both discrete and continuous latent representations in modern generative modelling.

ACKNOWLEDGMENTS

Denis Zavadski and Nikita Philip Tatsch are supported by the Konrad Zuse School of Excellence in Learning and Intelligent Systems (ELIZA) through the DAAD programme Konrad Zuse Schools of Excellence in Artificial Intelligence, sponsored by the Federal Ministry of Education and Research. The authors gratefully acknowledge the support by the Ministry of Science, Research and the Arts Baden-Württemberg (MWK) through bwHPC, SDS@hd and the German Research Foundation (DFG) through the grants INST 35/1597-1 FUGG and INST 35/1503-1 FUGG. The authors gratefully acknowledge the computing time made available to them on the high-performance computer at the NHR Center of TU Dresden under the DFG grant number 498181230.

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

Table 4: Training Detail Summary.

| Model | Training Data | Image Resolution | Training Steps | Batch Size | Learning Rate |
|---|---|---|---|---|---|
| PQGAN (All) | ImageNet | $256 \times 256$ | 1M | 20 | $4.5 \times 10^{-6}$ |
| PQGAN (Fine-Tuning) | ImageNet | $768 \times 768$ | 200k | 15 | $4.5 \times 10^{-6}$ |
| PQSD-Precise | Unsplash-Full | $384 \times 384$ | 20k | 32 | $8 \times 10^{-5}$ |
| | Unsplash-Full | $768 \times 768$ | 800k | 32 | $8 \times 10^{-5}$ |
| | Unsplash-Full | $768 \times 768$ | 200k | 32 | $2 \times 10^{-5}$ |
| PQSD-Quick | Unsplash-Full | $384 \times 384$ | 20k | 32 | $8 \times 10^{-5}$ |
| | Unsplash-Full | $768 \times 768$ | 800k | 32 | $8 \times 10^{-5}$ |
| | Unsplash-Full | $768 \times 768$ | 200k | 32 | $2 \times 10^{-5}$ |
| PQSD-HR | Unsplash-Full | $384 \times 384$ | 20k | 32 | $8 \times 10^{-5}$ |
| | Unsplash-Full | $768 \times 768$ | 400k | 32 | $8 \times 10^{-5}$ |
| | Unsplash-Full | $1024 \times 1024$ | 400k | 32 | $8 \times 10^{-5}$ |
| | Unsplash-Full | $1536 \times 1536$ | 200k | 32 | $2 \times 10^{-5}$ |

## A   TECHNICAL DETAILS

In the following, we describe additional training details for the training of all our PQGAN configurations and the adaptation of Stable Diffusion Rombach et al. (2022) to the PQGAN image representation space. Code, weights and models will be made available at GitHub for training and inference. We use the VQGAN Esser et al. (2021) framework, which is released under the MIT License. We modify StableDiffusion2.1 Rombach et al. (2022), which is released under the CreativeML Open RAIL++-M License.

### A.1   PQGAN TRAINING

All configurations of our PQGAN are trained within the VQGAN Esser et al. (2021) framework, using the original architecture and training schedule to preserve the purity of the comparison for the quantisation mechanism with product quantisation. All configurations are trained on ImageNet Russakovsky et al. (2015) images of size $256 \times 256$ for one million training steps with an overall batch size of 20 using 4×A100 GPUs. These models are used for all comparison with state-of-the-art. For downstream applications like latent image representation for generative diffusion models, we further fine-tune the best configurations according to PSNR, CMMD Jayasumana et al. (2024) and LPIPS Zhang et al. (2018). The best configuration for the used downscaling factors $F = 8$ and $F = 16$ are:

$$(F = 16) \qquad K = 128, \qquad d = 256, \qquad S = 128, \qquad (4)$$
$$(F = 8) \qquad K = 512, \qquad d = 128, \qquad S = 64, \qquad (5)$$

with codebook sizes $K$, latent dimension $d$ and number of sub-domains $S$. These two configurations are further fine-tuned for 200k training steps with an overall batch size of 15 on 3×H100 GPUs on ImageNet images with the increased resolution of $768 \times 768$.

### A.2   LATENT ADAPTATION TRAINING

We adapt Stable Diffusion 2.1 Rombach et al. (2022) (SDv2.1) to the latent image representations of our PQGAN using different compression factors $F = 8$ and $F = 16$. As training data, we use the Unsplash-Full dataset (Unsplash, 2025) consisting of 6.5 million high quality images. For the SDv2.1 generator, the input and output dimension is increased from 4 to the new latent dimension (128 for $F = 8$ and 256 for $F = 16$) by increasing the dimension of the first and last convolution layer. The new parameters are initialised by copying the previous pre-trained weights. First, only the input and output layers are trained while the rest of the diffusion model is kept frozen to not disturb the pre-trained weights of the generative model. This phase goes on for 20k training steps with images of resolution $512 \times 512$, an overall batch size of 32 over 4×H200 GPUs and a learning rate of $8 \times 10^{-5}$. We then unfreeze the model and train it for 800k more steps with a resolution of $768 \times 768$. For PQDS-Precise and PQSD-Quick, we keep training with $768 \times 768$ images for the remaining 200k steps, while we increase the resolution every 400k steps for our PQSD-HR model. A summary is shown in Table 4.

# B    CODEBOOK MATCHING SPEED

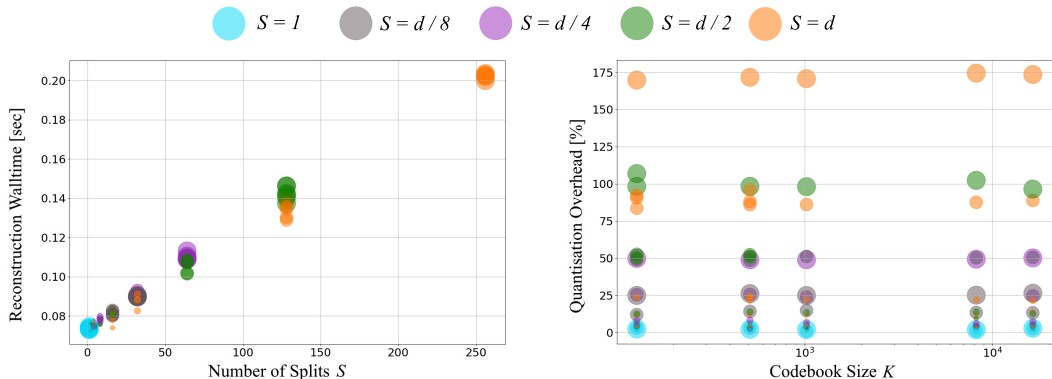

Figure 5: Reconstruction time and quantisation overhead comparison for product quantisation configurations.

In Figure 5, we show the walltime and the quantisation overhead for the reconstruction of an image for different product quantised configurations. We measure averaged times over 1000 reconstructions on a A100 GPU. Each marker represents a configuration, with colour indicating the number of subspaces $S$. Marker sizes reflect latent dimensionality $d$. The quantisation overhead Figure 5(right) is calculated as the ratio of the codebook matching time and the reconstruction time without quantisation. The codebook matching time depends mainly on the number of splits and is mostly independent of the actual codebook size up to 16 384 entries. The best performing configurations emerged to have $d = 128$ latent dimension with $S = 64$ subspaces and require therefore around 50% overhead for the codebook matching during the reconstruction process.

# C    PERFORMANCE TRENDS FOR PQ CONFIGURATIONS

In Figure 6, we show all trends for different PQ configurations with regards to PSNR, FID Heusel et al. (2017), CMMD Jayasumana et al. (2024) and LPIPS Zhang et al. (2018). Each marker represents a configuration, with colour indicating the number of subspaces $S$. In the left plot, marker size reflects codebook size $K$; in the middle and right plots, it reflects latent dimensionality $d$. All metrics follow the same trend reported for FID in the main paper: product quantised representations with high factorisation $S$ consistently achieve the strongest performance.

# D    GENERATIVE TRAINING FROM SCRATCH

Albeit the focus of the generative part of our work is the reparametrisation of a pre-trained generative model towards a new latent representation, we also train and show results for training a diffusion model from scratch on the PQGAN representation. We train a latent diffusion model (Rombach et al., 2022) with 400M parameters on ImageNet for 2M training steps and a batch size of 64 with the latent representation of StableDiffusion2.1 VAE and our PQGAN representations with the downsizing factors 8 and 16. The results are summarised in Table 5. Both PQGAN representation result in better downstream generation performance in terms of FID (Heusel et al., 2017).

# E    RECONSTRUCTION RESULTS

We reiterate the finding that current latent representations of in-production text-to-image generation models like StableDiffusion2.1 have clear limitations when it comes to high-resolution image representation. In Figure 7, we show additional examples of reconstructed images for our quantised PQGAN image representations that we use for the task of high resolution image generation. We

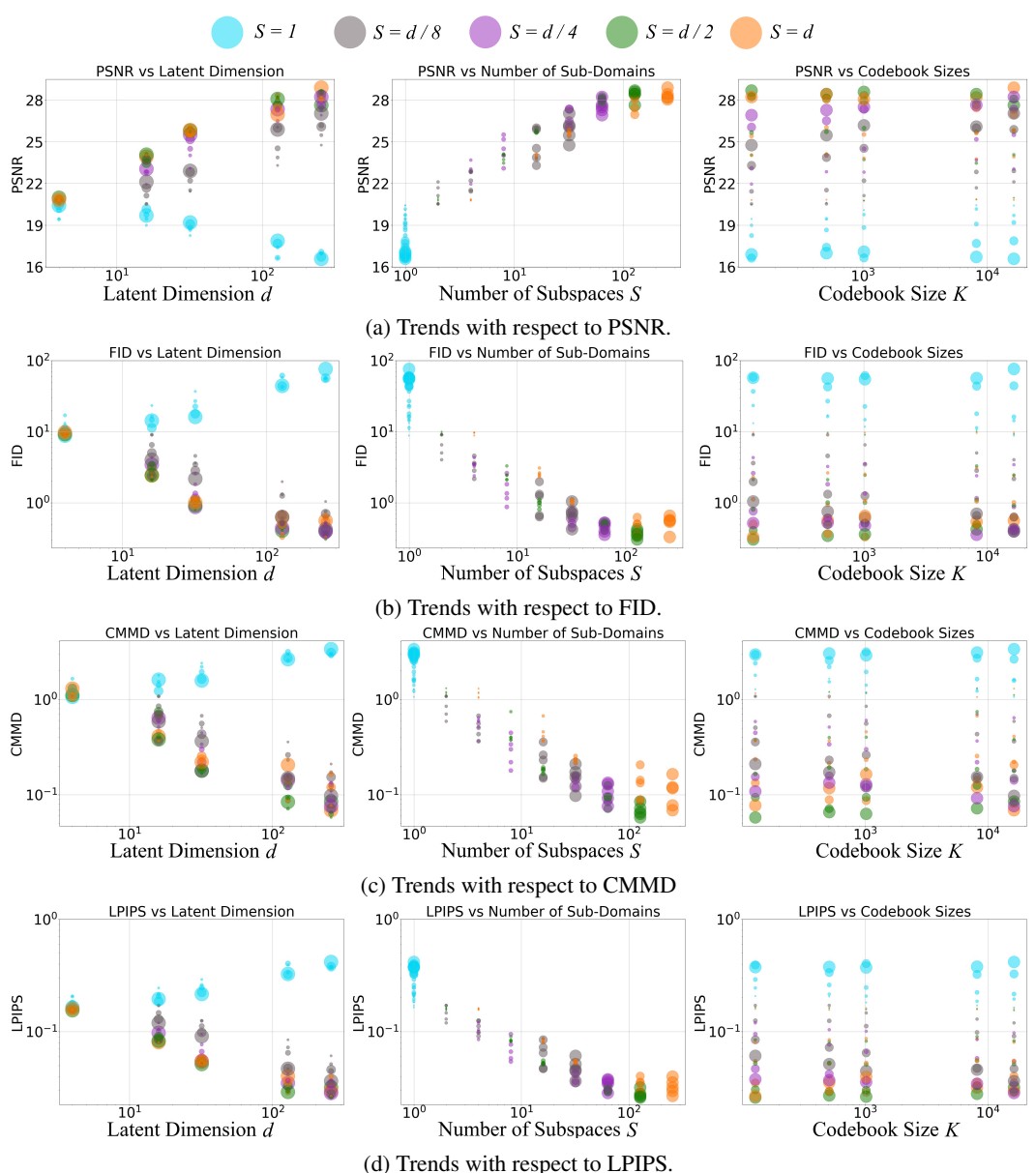

Figure 6: Quantitative analysis of product quantisation configurations. Performance is measured across latent dimensionality, number of subspaces $S$, and codebook size $K$. All metrics show the same trend with regard to configuration. Plots for FID, CMMD and LPIPS are on a log-log scale. Plots for PSNR are on a semi-log scale as PSNR is already in dB.

Table 5: Quantitative results for training latent diffusion model (Rombach et al., 2022) from scratch on the latent representations of the StableDiffusion2.1 VAE and PQGAN at different downsampling factors $F$ on ImageNet $256 \times 256$ (Russakovsky et al., 2015). The generative FID (gFID) is evaluated on 5k samples and the ImageNet validation set.

| Latent Model | F | Generator | Batch Size | Training Steps | rFID↓ | gFID↓ |
|---|---|---|---|---|---|---|
| SDv2.1-VAE | 8 | LDM | 64 | 2M | 0.75 | 27.21 |
| PQGAN (Ours) | 8 | LDM | 64 | 2M | 0.036 | 22.66 |
| | 16 | LDM | 64 | 2M | 0.41 | 21.08 |

again compare with the unconstrained reconstruction from the representation provided by SDv2.1-VAE in StableDiffusion2.1. The SDv2.1-VAE fails to faithfully preserve fine-grained image details, often producing blurred reconstructions or hallucinatory artefacts. In contrast, PQGAN with the same downsampling factor ($F = 8$) yields reconstructions that are visually near-indistinguishable from the original image. Even at $F = 16$, PQGAN retains a high level of structural and perceptual fidelity, exhibiting only minor deviations while still outperforming SDv2.1-VAE, despite using a spatially significantly more compressed representation.

# F ADDITIONAL QUALITATIVE RESULTS

We include additional qualitative examples from our PQSD variants to further reinforce the effectiveness of our latent representation for large-scale text-to-image diffusion models. Figure 8 shows samples from our PQSD-Precise model, using the $F = 8$ compressed space, for resolutions of $768 \times 768$ and $768 \times 1536$. Figures 9, 10 show samples for our PQSD-HR and PQSD-Quick adapted models, using the $F = 16$ compression space, for resolutions of $768 \times 768$ to $1536 \times 3072$.

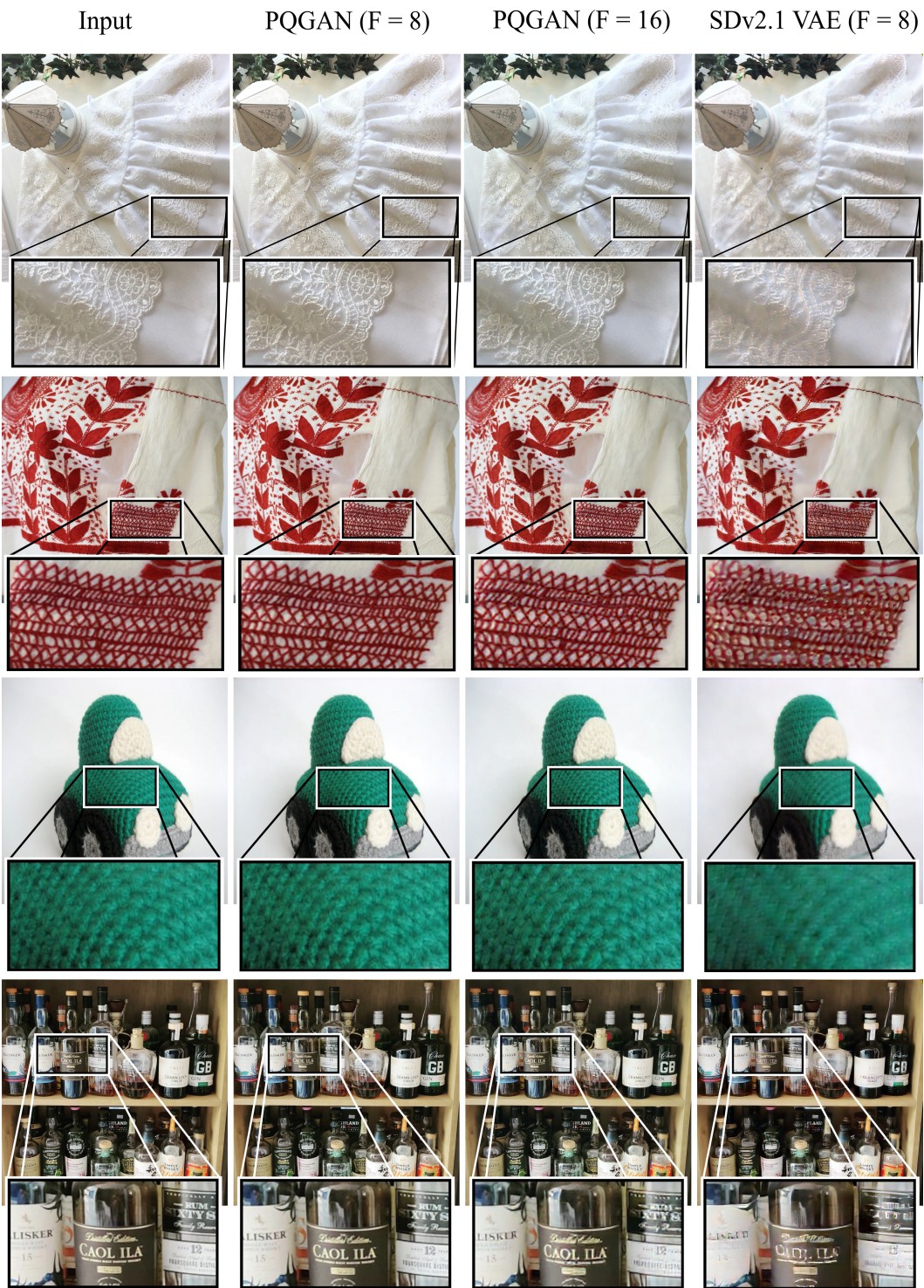

Figure 7: Reconstruction results for our quantised PQGANs vs. the continuous VAE in StableDiffusion2.1 Rombach et al. (2022). Even with stronger downsizing $F$, the reconstructions from our representation space show higher fidelity. The representation space of StableDiffusion2.1 has trouble preserving information for high resolutions ($1536 \times 1536$, top image) but also loses information for its native resolution ($768 \times 768$, rest).

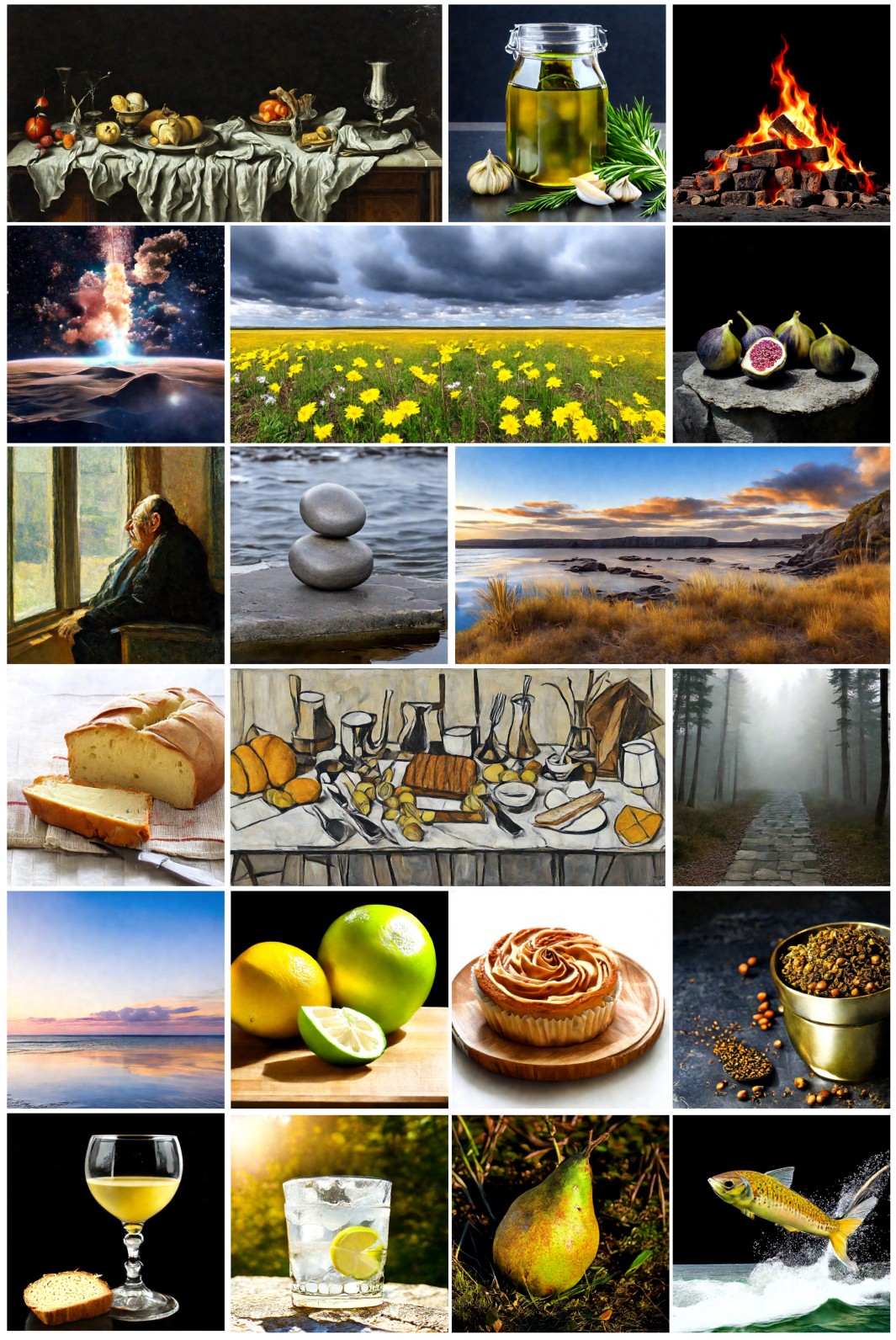

Figure 8: Additional samples from our PQSD-Precise model adapted to operate on our $F = 8$ product quantised image representation space. Samples have the resolutions $768 \times 768$ and $768 \times 1536$.

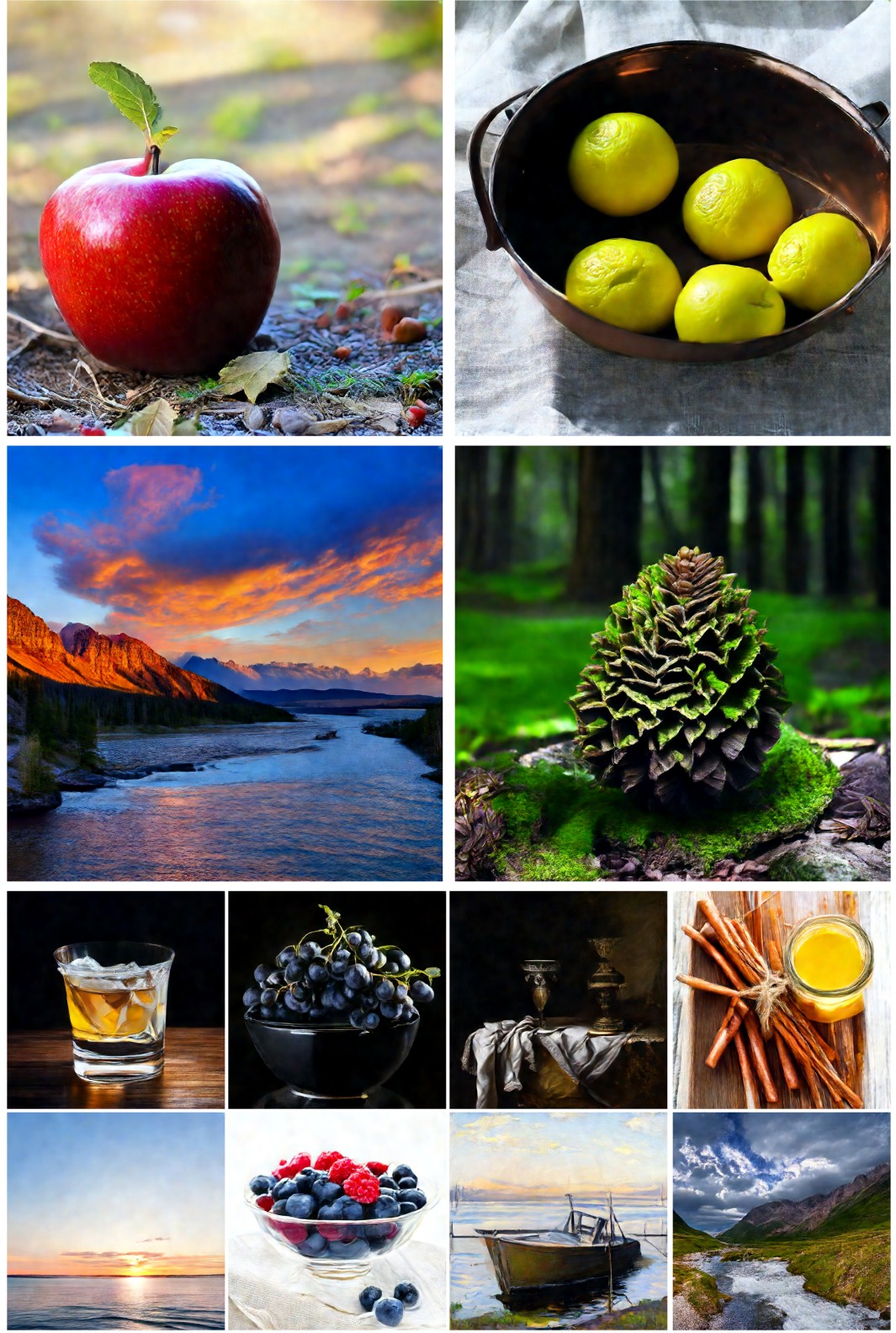

Figure 9: Additional samples from our PQSD-HR model (first two rows) and our PQSD-Quick model (last two rows) adapted to operate on our $F = 16$ product quantised image representation space. Samples have the resolutions $768 \times 768$ and $1536 \times 1536$.

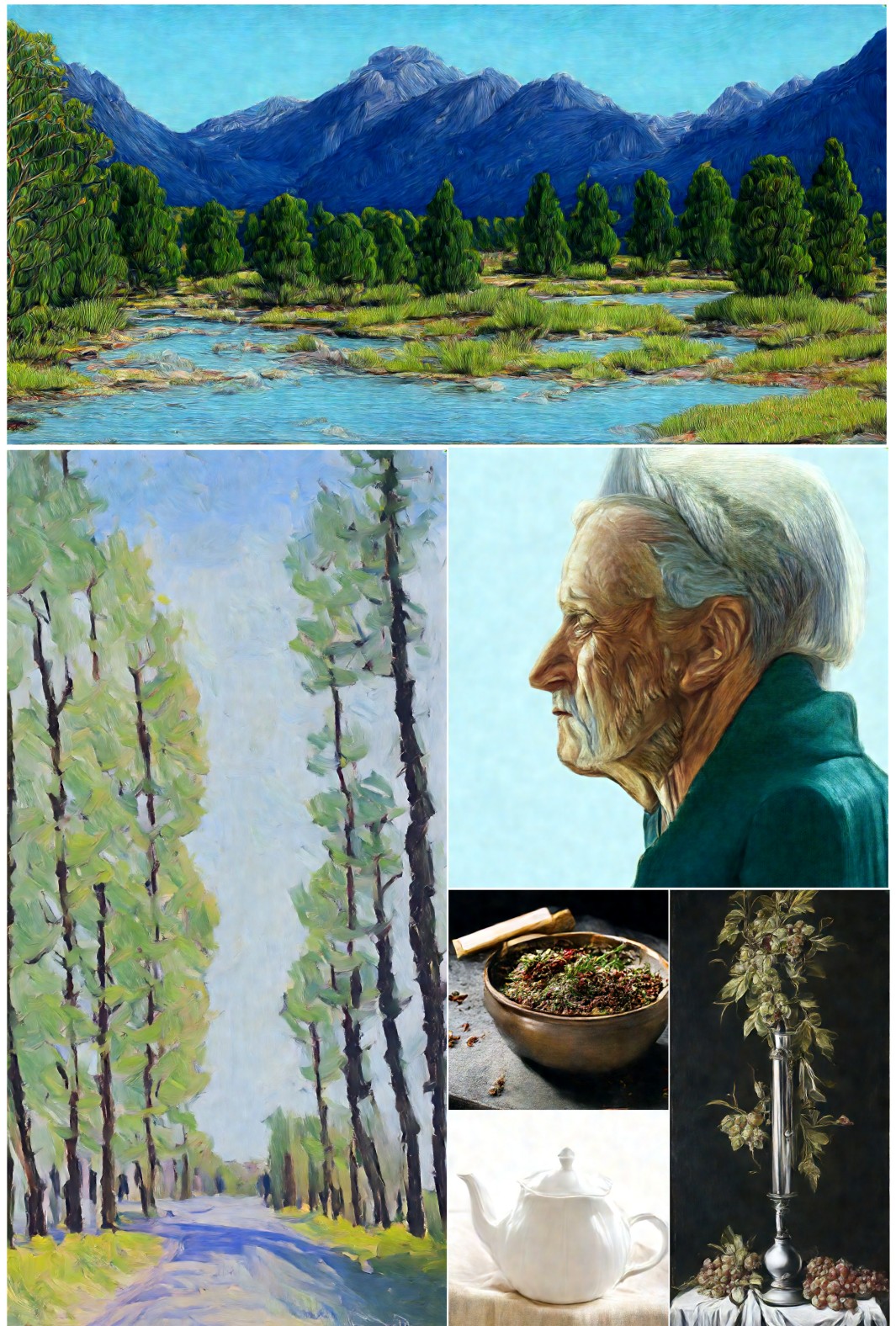

Figure 10: Additional samples from our PQSD-HR model (top three images) and our PQSD-Quick model (bottom-right three images) adapted to operate on our $F = 16$ product quantised image representation space. Samples have the resolutions $768 \times 768$, $1536 \times 1536$, $1536 \times 768$, $3072 \times 1536$ and $3072 \times 1536$.

