# OpenReview forum: "PQGAN: Product-Quantised Image Representation for High-Quality Image Synthesis"
_ICLR.cc/2026/Conference — ICLR 2026 Poster_

### Official Review · Reviewer_rVnn · 2025-10-27

**Soundness:** 3
**Presentation:** 4
**Contribution:** 3
**Rating:** 8
**Confidence:** 4

**Summary:**

This paper introduces PQGAN, a novel quantised autoencoder that uses product quantisation (PQ) instead of the traditional vector quantisation (VQ) used in models like VQGAN. The simple yet effective idea is to splits the vector into S subspaces, each quantized separately. This allow a codebook which is exponentially bigger than the VQ codebook, depending on the dimension of the subspaces (with one subspace only, PQ reduces to VQ).

The motivation behind PQGAN is that standard VQ suffers from training sparsity, codebook collapse, and redundancy: PQGAN addresses this by factorising each latent vector into subspaces, quantising each independently.

On ImageNet 256×256, PQGAN outperforms the  baselines and achieves high fidelity with small codebooks (K = 128–512).

The paper carries on a detailed analysis on codebook usage (in terms of Perplexity and Entropy) and metrics are evaluated against size of the codebook, number of subspaces and latent embedding size.

Finally, PQGAN was integrated into Stable Diffusion 2.1, in three variants improving generation results.

**Strengths:**

- Novelty: despite being simple, the idea is novel and effective.

- The paper conducts a systematic analysis of how embedding dimension, number of subspaces, and codebook size interact, covering the full spectrum between scalar and vector quantisation.

- Extensive experiments on ImageNet, FFHQ, and LSUN establish both quantitative and qualitative superiority. Reported improvements are substantial and consistent.

- Integration with Stable Diffusion is a useful addition to the method and is carefully validated, showing also computational benefit.

- I have appreciated the analysis of codebook utilisation, with entropy and perplexity evaulations. Codebook utilization is an issue in standard VQ-VAEs

- The paper is clearly written and well-structured. The presentation of results is well-designed

**Weaknesses:**

I think the main weakness of the paper is the substantial lack of theoretical foundation: PQ seems to be a well-established technique in signal processing and vector encoding [1], and its application to autoencoders has appeared in other context contexts (e.g., El-Nouby et al., 2023; Mentzer et al., 2020) as highlighted in the paper itself.

A deeper theoretical argument for why PQ improves latent scalability and codebook utilization (rather than relying only on empirical evidence) may strengthen the paper.

The paper also claims that PQ and VQ behave “in opposite ways when scaling the embedding dimension,” but this is again presented empirically without theoretical insight.

[1] Product Quantization for Nearest Neighbor Search, Jegou et al. - TPAMI 2011

**Questions:**

Please refer to the "Weakness".

I think the paper would benefit from more insights (possibly theoretical) on the reasons why PO obtains such successful results.

---

> ### Author Response · Authors · 2025-11-18
> **Rebuttal by Authors**
>
> Thank you for the review. We will address the weaknesses, as asked in the questions.
>
>
> >*"I think the main weakness of the paper is the substantial lack of theoretical foundation: PQ seems to be a well-established technique in signal processing and vector encoding, and its application to autoencoders has appeared in other context contexts (e.g., El-Nouby et al., 2023; Mentzer et al., 2020) as highlighted in the paper itself.*
> >
> >*A deeper theoretical argument for why PQ improves latent scalability and codebook utilization (rather than relying only on empirical evidence) may strengthen the paper.*
> >
> >*The paper also claims that PQ and VQ behave “in opposite ways when scaling the embedding dimension,” but this is again presented empirically without theoretical insight."*
>
>  We address the above questions jointly.  We attribute the divergent behaviour of PQ and VQ, as well as PQ’s improved scalability and codebook utilisation, primarily to two interrelated phenomena. We will expand on both in a revised version:
>
> 1. **Training signal sparsity and sample efficiency.**
> Vector quantisation operates on the full entangled $d$-dimensional latent space, and learning a meaningful codebook requires many training samples to sufficiently populate that space. To achieve a covering resolution $\epsilon$, the required number of samples grows as $\mathcal{O}((1/\epsilon)^d)$, leading to severe data sparsity in high dimensions. This impairs codebook learning, as each centroid receives little supervision. In contrast, PQGAN factorises the space into $S=d/2$ subspaces of dimension 2, each of which can be learned independently with only $\mathcal{O}((1/\epsilon)^2)$ samples to achieve the same latent coverage resolution of $\epsilon$, regardless of the latent dimension . This keeps the effective sample density constant as $d$ increases, allowing PQ to scale latent dimensionality without sacrificing codebook fidelity.
>
> 2. **Quantisation error scaling.** Classical quantisation theory [1] shows that mean squared quantisation error scales as $\mathcal{O}(K^{-2/d})$ in $d$-dimensional space with $K$ centroids. For VQ, this implies that increasing latent dimension requires exponential growth in codebook size to maintain the same quantisation error, which is an impractical trade-off. PQ sidesteps this by operating in fixed low-dimensional subspaces (with $S = d/2$), where the quantisation error remain constant $\mathcal{O}(K^{-2/2})$, independent of total latent dimension. This allows PQ to increase total representational capacity (via higher $d$) without degrading quantisation error or requiring larger codebooks.
>
> Together, these principles explain the key empirical trend: VQ degrades with increasing latent dimensionality, while PQ improves. These results are not merely empirical observations, they are a direct consequence of how VQ and PQ differ in their scaling properties with respect to training signal efficiency and quantisation error. We will add this discussion after L203 in the main text where we mention training signal sparsity.
>
> [1] Zador, Paul. "Asymptotic quantization error of continuous signals and the quantization dimension." IEEE Transactions on Information Theory 28.2 (1982): 139-149.

---

> > ### Comment · Reviewer_rVnn · 2025-11-24
> >
> > Thanks a lot for the answer, I think that is a great addition to the paper.

---

### Official Review · Reviewer_Jhoa · 2025-10-29

**Soundness:** 2
**Presentation:** 2
**Contribution:** 2
**Rating:** 2
**Confidence:** 5

**Summary:**

This paper introduces PQGAN, which incorporates product quantisation (PQ) into the VQ-VAE framework. PQGAN achieves state-of-the-art reconstruction performance, significantly improving metrics such as PSNR, FID, LPIPS, and CMMD compared to existing methods. Furthermore, the paper demonstrates that PQGAN can be seamlessly integrated with pre-trained diffusion models, resulting in faster or higher-resolution image generation.

**Strengths:**

- PQGAN demonstrates significant improvements in image reconstruction compared to state-of-the-art methods.

**Weaknesses:**

- The novelty of the overall approach is very limited. PQ has been used for some time in the field of VQ-VAE, e.g., UniTok [a]. This paper is more like a technical report than a research paper, and it proposes few new ideas or inspirations.

- The proposed latent adaptation is also very straightforward; reducing the spatial resolution of the VAE and increasing the number of channels are common techniques for reducing computation when training diffusion models. This application does not necessarily demonstrate that PQ brings any additional benefits.

- The comparison with previous works is unfair, as this method uses significantly more tokens than other VQ-VAEs.

[a] Unitok: A unified tokenizer for visual generation and understanding. C Ma, Y Jiang, J Wu, J Yang, X Yu, Z Yuan, B Peng, X Qi. arXiv preprint arXiv:2502.20321

**Questions:**

- The authors are suggested to highlight the contribution of this work.

---

> ### Author Response · Authors · 2025-11-18
> **Rebuttal by Authors**
>
> Thank you for the review. We address the concerns as follows.
>
> **Weaknesses**
>
> >*"The novelty of the overall approach is very limited. PQ has been used for some time in the field of VQ-VAE, e.g., UniTok."*
>
> We respectively disagree with this assessment, and give four arguments:
>
> 1. To avoid misunderstandings, our work specifically addresses quantised image representations for diffusion and flow-based models (see e.g., L87–93, L107, L151, L485). In this context, the number of tokens is not a limiting factor, which allows us to fully exploit the scalability of product quantisation (PQ). There exists a separate line of work exploring Multi-Channel Vector Quantisation (MCQ), where token count is critical, particularly for autoregressive generation. Among these, Mo-VQGAN is a representative baseline, which we directly compare to and substantially outperform. UniTok is another MCQ-style method, though it does not reference Mo-VQGAN. We will make this aspect more clear in the main text.
>
> 2. UniTok was at the time of submission (ICLR deadline was 24.9) a not peer-reviewed ArXiv article. According to ICLR guidelines we do not have to refer to such articles.
>
> 3. We are not aware of any other work that employs product quantisation to its full potential in the context of image representation for diffusion or flow-based models. Could you please provide further references?
>
> 4. Finally, UniTok does conceptionally not apply to our comparison for the following reason. As stated several times in the paper (L161, L225, L361, L673), we compare our work to state-of-the-art quantisation approaches, while keeping a controlled setup within the common VQGAN framework. This explicitly allows us to assess the effect of quantisation methods within one uniform framework, rather than having potentially misleading results across additionally different models and training pipelines. As a brief comparison of set-ups, In contrast to the VQGAN framework, UniTok uses the completely different and several times larger ViTamin (2) architecture and trains on a larger dataset of a higher quality.
>
>
> >*"The comparison with previous works is unfair, as this method uses significantly more tokens than other VQ-VAEs."*
>
> We respectfully disagree. As mentioned above, the submission is exploring quantised image representation for Diffusion and Flow-based models. In this context, tokens do not matter, which allows us to utilise and evaluate PQ at its full potential. The only relevant parameters in the comparison with quantised models are the latent resolution, the latent dimension and the codebook size. Our PQGAN outperforms competitors for the same and even less favourable settings.
>
> (2) Jieneng Chen, Qihang Yu, Xiaohui Shen, Alan Yuille, and Liang-Chieh Chen. Vitamin:Designing scalable vision models in the vision-language era. In Proceedings of the IEEE/CVF Conference on Computer Vision and Pattern Recognition, pages 12954–12966, 2024.

---

> > ### Comment · Reviewer_Jhoa · 2025-11-25
> >
> > Thank you for your detailed response. I appreciate the clarifications regarding your positioning of PQGAN within the context of quantized representations for diffusion and flow-based models. However, I would like to further elaborate on my concerns:
> >
> > 1. I remain unconvinced that quantized image representations are essential or uniquely beneficial for diffusion and flow-based models. While quantization can facilitate certain trade-offs in memory and computation, there is no clear evidence presented that it is necessary for effective diffusion/flow modeling, or that it offers distinct advantages over continuous representations in this context.
> >
> > 2. I still find the experimental comparison with VAE-based baselines to be unfair. In your experiments, PQGAN employs a significantly larger latent dimension (e.g., 128 vs. 4 in competing VAEs). This substantially increases the model’s capacity and expressiveness, making direct performance comparisons unfair. A more meaningful evaluation would control for latent dimensionality, backbone structure, and training data, thereby comparing the effect of the quantization method (PQ vs. KL) itself.
> >
> > 3. While I appreciate your points regarding UniTok, I would like to highlight that there are several other works that employ product quantization within the VQ-VAE framework, such as FQGAN [a]. These methods also leverage PQ for visual generation tasks and should be discussed to provide a comprehensive review of related literature.
> >
> > [a] Bai, Zechen, et al. "Factorized visual tokenization and generation." arXiv preprint arXiv:2411.16681 (2024).

---

> > > ### Author Response · Authors · 2025-11-26
> > > **Rebuttal Answer by the Authors**
> > >
> > > Thank you for the detailed answer. We are glad to see that we resolved the misunderstandings with respect to the point that the number of tokens is not a limiting factor in our scenario.
> > >
> > > We provide our response below:
> > >
> > > >*"I remain unconvinced that quantised image representations are essential or uniquely beneficial for diffusion and flow-based models. While quantisation can facilitate certain trade-offs in memory and computation, there is no clear evidence presented that it is necessary for effective diffusion/flow modeling, or that it offers distinct advantages over continuous representations in this context."*
> > >
> > > Let us first say how we understand your question. In your initial review you said: "The comparison with previous works is unfair, as this method uses significantly more tokens than other VQ-VAEs". In this question you ask about "no clear evidence presented that it is necessary for effective diffusion/flow modelling". These are different questions, and hence we conclude that you are now convinced by the main result of our work, i.e. showing the benefit of PQ over VQ and published VAE (see table 1 and 2). Please let us know if you are still unconvinced.
> > >
> > > Let us now answer your concern with respect to lack of evidence. We show direct advantages of applying our representation to a common pre-trained generative model (here SDv2.1) in table 3 and Section 4.5 (L467-L474) and Table 5. To summarise, in table 5 we show that the rFID score and gFID score are considerable better for training on our PQGAN representation (compared to SDv2.1-VAE) while having the same (or more) number of samples per second and the same (or smaller) memory consumption (as seen in table 3).
> > >
> > >
> > > >*"I still find the experimental comparison with VAE-based baselines to be unfair. In your experiments, PQGAN employs a significantly larger latent dimension (e.g., 128 vs. 4 in competing VAEs). This substantially increases the model’s capacity and expressiveness, making direct performance comparisons unfair. A more meaningful evaluation would control for latent dimensionality, backbone structure, and training data, thereby comparing the effect of the quantisation method (PQ vs. KL) itself."*
> > >
> > > First, we want to emphasise that the the main focus of our work is about quantised representations and therefore we mainly compare with quantisation methods. Secondly, we understand your question as follows. We assume that you would have liked us to take the models (see table 1): SDv2.1 VAE, LDM VAE, SDXL VAE and change for all of them the dimension $d$ from 4 to e.g. 128. We did not do that for two reasons. Firstly, these are the published models which we do not want to change. If we start to adapt the hyperparamater $d$ then maybe other parameters of these models needs also adjustments to give good performance. We assume that the authors of the respective published models did their best to get the best results. It is not our task to try to improve published  work. Second, we are not sure if we are able to reproduce their model for various reasons such as computational power (e.g. we have batch size 20 while Stable Diffusion XL VAE has a batch size of 256).
> > >
> > >
> > > >*"While I appreciate your points regarding UniTok, I would like to highlight that there are several other works that employ product quantisation within the VQ-VAE framework, such as FQGAN [a]. These methods also leverage PQ for visual generation tasks and should be discussed to provide a comprehensive review of related literature."*
> > >
> > > Thank you for the pointer. However, as said in our previous reply, according to ICLR guidelines we do not have to discuss ArXiv articles that have not been peer-reviewed. However, we would still like to point out the differences to our work for completeness. FQGAN uses additional adapters to map to 2 or 3 sub codebooks from which latent pixels are constructed. Furthermore, FQGAN relies on additional losses to guide the codebook learning, like the explicit disentangling of the codebooks, but more importantly, the guidance of single codebooks with different pre-trained networks like CLIP and DINOv2. Thus, the approach also operates at the very low end of product quantisation in terms of codebook splitting, and it appears unable to scale the number of sub-codebooks because of the manual assignment of (sub-codebook, pre-trained-encoder) pairs. Therefore, with respect to quantisation, it still can be categorised as a form of engineered Mo-VQGAN. PQGAN still strongly outperforms the method in question. In the following table, we compare the numbers from the FQGAN work with ours for the reconstruction of ImageNet $256\times256$ images.
> > >
> > >
> > > | | **F = 16** |        | **F = 8** |        |
> > > |--------|------------|--------|-----------|--------|
> > > |**Method**|**rFID ↓**|**PSNR ↑**|**rFID ↓**| **PSNR ↑**|
> > > | FQGAN  | 0.76       | 22.7   | 0.24      | 27.6   |
> > > | PQGAN (ours) | **0.41** | **28.3** | **0.036** | **37.4** |

---

> > > > ### Comment · Reviewer_Jhoa · 2025-11-27
> > > >
> > > > > In your initial review, you said: "The comparison with previous works is unfair, as this method uses significantly more tokens than other VQ-VAEs." In this question, you ask about "no clear evidence presented that it is necessary for effective diffusion/flow modeling." These are different questions, and hence we conclude that you are now convinced by the main result of our work, i.e., showing the benefit of PQ over VQ and published VAE (see Table 1 and 2). Please let us know if you are still unconvinced.
> > > >
> > > > I am not convinced. The core issue remains: your method uses significantly more tokens than other VQ-VAEs. For quantized image representation, especially in the context of **reconstruction**, the number of tokens is a crucial metric, more tokens generally enable better reconstruction, regardless of the quantization method. Latent resolution alone is not a sufficient basis for fair comparison. Therefore, the comparisons with VQ in Table 1 (and similarly with FQGAN in your rebuttal) are not meaningful, as they do not control for the number of tokens. The improved results may simply reflect increased representational capacity due to more tokens, rather than a true advantage of PQGAN as a quantization method.
> > > >
> > > >
> > > > >Let us now answer your concern with respect to lack of evidence. We show direct advantages of applying our representation to a common pre-trained generative model (here SDv2.1) in Table 3 and Section 4.5 (L467–L474) and Table 5. To summarise, in Table 5 we show that the rFID score and gFID score are considerably better for training on our PQGAN representation (compared to SDv2.1-VAE) while having the same (or more) number of samples per second and the same (or smaller) memory consumption (as seen in Table 3).
> > > >
> > > > The experiments in Table 3 primarily show the benefits of increasing latent dimensionality, not the effect of PQGAN itself. It is likely that any VAE with a similar latent dimension would achieve comparable results. This is the main reason I have repeatedly raised concerns about experimental fairness, but the responses so far have not directly addressed this point. Furthermore, the experiments in Table 5 are problematic because the training data differs between compared models, making it impossible to isolate the contribution of PQGAN. Overall, the experiments demonstrate that diffusion models can be adapted to PQ-based representations, but do not provide clear evidence for the unique effectiveness or necessity of quantized image representations (or PQGAN specifically) for diffusion and flow-based models.
> > > >
> > > > > We want to emphasise that the main focus of our work is about quantised representations and therefore we mainly compare with quantisation methods.
> > > >
> > > > As also noted by other reviewers (Nh49 and rVnn), product quantisation (PQ) is a well-established technique, and its use for quantised representations is not novel. Simply applying PQ within this context does not represent a significant research contribution.

---

> > > > > ### Author Response · Authors · 2025-12-02
> > > > > **Rebuttal Answer by Authors**
> > > > >
> > > > > Thank you for your response. We address the raised points below.
> > > > >
> > > > > >"*I am not convinced. The core issue remains:
> > > > >     your method uses significantly more tokens than other VQ-VAEs. For quantized image representation, especially in the context of **reconstruction**, the number of tokens is a crucial metric, more tokens generally enable better reconstruction, regardless of the quantization method. Latent resolution alone is not a sufficient basis for fair comparison. Therefore, the comparisons with VQ in Table 1 (and similarly with FQGAN in your rebuttal) are not meaningful, as they do not control for the number of tokens.
> > > > >     The improved results may simply reflect increased representational capacity due to more tokens, rather than a true advantage of PQGAN as a quantization method.*"
> > > > >
> > > > > 1. What you seem to refer to is **compression** not **reconstruction**. Please keep in mind, that all metrics need to be reasonable for the task at hand. As we mention in the submission (L81ff, L119-L122, L147-L150), for tasks like compression, limited tokens and indices certainly are crucial. However, as we explain in the paper (L85-L90, L151ff) and also already in the rebuttal before, we consider the application of representations for diffusion and flow-based models. Tokens are completely irrelevant in this context.
> > > > >
> > > > > 2. According to your remark about fairness, this would mean that comparisons of standard VQGAN with methods on the basis of Mo-VQGAN that use multi channel quantisation and RQ-VAE that use residual quantisation would be all unfair, since all of them increase tokenisation. However, again the crucial aspect is what these methods are eventually used for. Please keep in mind that all methods have tradeoffs. The compared quantisation approaches have a considerably lower reconstruction quality, but can be applied for autoregressive models. In contrast, we have highly increased representation quality with smaller dimensions and/or codebooks sizes as compared quantised approaches. The tradeoff for our method is (as stated in L83f) that the representation can not be applied right away to autoregressive models but works well for diffusion and flow-based approaches.
> > > > >
> > > > > >"*The experiments in Table 3 primarily show the benefits of increasing latent dimensionality, not the effect of PQGAN itself. It is likely that any VAE with a similar latent dimension would achieve comparable results. This is the main reason I have repeatedly raised concerns about experimental fairness, but the responses so far have not directly addressed this point.
> > > > >     Furthermore, the experiments in Table 5 are problematic because the training data differs between compared models, making it impossible to isolate the contribution of PQGAN. Overall, the experiments demonstrate that diffusion models can be adapted to PQ-based representations, but do not provide clear evidence for the unique effectiveness or necessity of quantized image representations (or PQGAN specifically) for diffusion and flow-based models.*"
> > > > >
> > > > > 1. Respectfully, this is not correct. The whole reason of Table 5 was to show the effect in a controlled training setup for the downstream task, therefore the training data is the same.
> > > > >
> > > > > 2. The concerns raised appear to be based more on conjecture than on specific factual evidence. All our comparisons are conducted using the official weights of widely established methods. The effectiveness of our proposed representation is evaluated in the generative setting through direct comparison to existing models and is consistently supported by quantitative results.
> > > > >
> > > > > 3. Regarding fairness and dimensionality, Table 1 shows that other quantisation methods, even at the same dimensionality, do not achieve the reconstruction performance gains observed with PQGAN. As discussed in the paper and elaborated in the rebuttal (e.g., in response to Reviewer rVnn), this stems from the fundamental differences in how VQ and PQ scale with latent dimensionality, particularly with respect to training signal density and quantisation efficiency.
> > > > >
> > > > > 3. As we state in our contributions, we propose the PQGAN representation, show its increased reconstruction quality, show its applicability to pre-trained diffusion models and the direct benefits. Please note, we never claim that our representation is the only one possible. What we do however, is to show its proficiency in comparison to established models.

---

> ### Author Response · Authors · 2025-12-02
> **Novelty Remark - Answer by Authors**
>
> We would like to address the novelty remark below:
>
> >"*As also noted by other reviewers (Nh49 and rVnn), product quantisation (PQ) is a well-established technique, and its use for quantised representations is not novel. Simply applying PQ within this context does not represent a significant research contribution.*"
>
> We clearly acknowledge that product quantisation is an existing technique (in fact, we reference it explicitly in the very first sentence of the abstract) and we discuss prior applications and their limitations in the related work section. Other reviewers have suggested adding a standard citation, which we agree with and will include. Notably, Reviewer rVnn explicitly stated: “Novelty: despite being simple, the idea is novel and effective.” While we understand that reviewers may hold differing views, we respectfully ask that the evaluation remains grounded in the factual content of the submission.
>
> Regarding the comment that “simply applying PQ within this context does not represent a significant research contribution,” we refer to the four stated contributions of the paper.

---

### Official Review · Reviewer_gjYk · 2025-11-01

**Soundness:** 3
**Presentation:** 3
**Contribution:** 3
**Rating:** 6
**Confidence:** 3

**Summary:**

This paper introduces PQGAN, a novel way to quantize the latents of the auto encoders using product quantized learning. This method factorizes the high dimensional latent channel into many smaller, independent subspaces, each quantized with their own codebook. This method improve the reconstruction quality of the VAE and achieves a state-of-the-art 37.4 dB PSNR. The authors argue that the spatial resolution is the main bottleneck and through their method, they can operate at a lower spatial resolution but with a much higher channel dimension. This enables the method to generate larger resolution images or get speed up upto 4x.

**Strengths:**

1)  PQGAN achieves very high reconstruction fidelity, 37.4 dB PSNR, which is higher than 25.3 dB of the standard Stable Diffusion VAE and other methods.

2) The paper demonstrates a novel finding the product quantization improves the reconstruction quality in VAEs

3) The method can either double the output image resolution or achieving a 4x generation speedup at the same resolution, This is achieved with the same cost.

**Weaknesses:**

1) The paper fails to provide  quantitative comparison (e.g., FID, CLIP Score) between the generations of its adapted PQSD model and the original Stable Diffusion.

2) The independent learning of codebooks might not learn complex correlations as the number of subspaces increase. In the limit it is as if sampling independently from each dimension. The tend in the paper also shows that.

3) Since multiple indices are associated with the same pixel, it is incompatible with the autoregressive models. Only Flow based models and diffusion models can benefit from this.

**Questions:**

1) The paper does not provide quantitative generative metrics like FID or CLIP score to compare their with other methods. This is the main limitation and it would be good to see these scores to validate the claims in the paper. Why are these evaluations not in the paper?

2. Discuss more about the tradeoff in weakness 2). What is the cut-off?

3. 50% increase in the inference cost seems high. What is the increase in the training time?

---

> ### Author Response · Authors · 2025-11-18
> **Rebuttal by Authors**
>
> Thank you for the review. Since the weaknesses are reflected in the questions, we will only address the questions.
>
> **Questions**
>
> >*(1) "The paper does not provide quantitative generative metrics like FID or CLIP score to compare their with other methods. This is the main limitation and it would be good to see these scores to validate the claims in the paper. Why are these evaluations not in the paper?"*
>
> Metrics like FID and CLIP Score are highly sensitive to the choice of training data, especially when comparing across different image domains. Our PQSD models are trained on the Unsplash dataset because of the increased availability of high-resolution images, whereas Stable Diffusion uses LAION. This domain shift makes direct comparisons potentially misleading, as these metrics may reflect dataset biases rather than model capabilities.
>
> To account for this, we present in L434-L426 and Appendix D (Table 5) a comparison where we re-train the latent diffusion model from scratch on the same PQGAN and Stable Diffusion VAE representations, using the same training data. In this controlled setting, the PQGAN representations yield significantly lower FID, validating the effectiveness of the PQ-based latent structure.
>
> >*(2) "The independent learning of codebooks might not learn complex correlations as the number of subspaces increase. In the limit it is as if sampling independently from each dimension. The trend in the paper also shows that. What is the cut-off?"*
>
> We see two main reasons for the trade-off
>
> 1. Unrestricted dependency of codebooks. The grouping of codebook dimensions in our PQGAN is not pre-conditioned, meaning that high-dimensional codebooks can have uncorrelated dimensions which can obstruct training. By choosing smaller dimensions for each codebook, we decrease the probability of having uncorrelated dimensions in one codebook and transfer the cross-correlation learning from the codebook to the encoder/decoder. This means that correlations suggested by the encoder are retained by a more faithful quantisation of PQ codebooks, while in the VQ case, they could be overwritten for several dimensions. However, having some cross correlations within the codebooks shows to be empirically advantageous and the cutoff occurs consistently at $S = d/2$ splits, so with a per codebook dimension of 2.
>
> 2. In VQ, all dimensions are learned jointly and therefore have also uncorrelated dimensions by construction. Therefore, VQ would have to store uncorrelated dimensions as redundant codebook entries, inflating the needed codebook size exponentially, as mentioned in the paper L53-L73. We will expand the explanation in the paper to improve clarity.
>
>
> >*(3) "50\% increase in the inference cost seems high. What is the increase in the training time?"*
>
> The 50\% inference cost of the PQGAN only affects its reconstructions.  During training, the compute requirement shift towards gradient computation. As such, Product quantisation introduces an overhead of 7.2\% in comparison with full vector quantisation. Also, in the generative case, the increased inference of PQGAN remains negligible as shown in the inference time comparison in Table 3.

---

### Official Review · Reviewer_Nh49 · 2025-11-03

**Soundness:** 3
**Presentation:** 3
**Contribution:** 3
**Rating:** 6
**Confidence:** 2

**Summary:**

This paper introduces PQGAN, a novel quantised image autoencoder that utilizes Product Quantisation (PQ) to achieve state-of-the-art fidelity for latent image representations, especially for high-quality image synthesis. PQGAN is integrated into pre-trained diffusion models, like Stable Diffusionl, which provides significantly faster and more compute-efficient generation, and improve the output resolution.
The paper also reports difference behavior of VQ and PQ with respect to embedding dimension.

**Strengths:**

PQGAN surpasses both existing quantisation methods and continuous autoencoders in reconstruction quality

**Weaknesses:**

This work is not the first to consider PQ (or RQ) for image compression, which is mentioned in the paper, yet the phrasing is still misleading in some places (abstract). That's being said, the method of this paper is much better. It would be worth ablating and analyzing more why.

For the mundane reader, it would be worth citing the paper that has introduced/popularized product quantization ("Product quantization for nearest neighbor search").

typo L329: benifits

**Questions:**

From Table 1, I understand that the big improvement in PSNR comes from using a resolution of 32x32, while the 16x16 patch size is only offering a PSNR of 28.3.
Have you tried larger patch sizes?

Similarly, RVQ is not evaluated in the context of compression, and only reported with patch size of 8x8. So I wonder if the better performance of PQ in this context is simply due to a better hyper-parameter choice of the patch size. Have you tried replacing PQ by a RVQ?

---

> ### Author Response · Authors · 2025-11-18
> **Rebuttal by Authors**
>
> Thank you for the review. We address the raised points as follows.
>
>
> **Weaknesses**
> > *"the method of this paper is much better. It would be worth ablating and analyzing more why."*
>
> Thank you for the suggestion. We will include a paragraph with additional mathematical intuition for the improved performance of PQ. We attribute the empirical advantage of PQ to a difference in training signal density between PQ and VQ. In VQ, each training example contributes one sample to a high-dimensional, entangled latent space. To cover this space at a fixed granularity $\epsilon$, the required number of samples grows as $\mathcal{O}((\frac{1}{\epsilon})^d)$, see [1]. This leads to severe sparsity in the coverage of the learned latent space for VQ. In contrast, PQ factorises the latent space into $S$ subspaces of dimension $d/S$. Since each subspace is trained independently, only $\mathcal{O}((\frac{1}{\epsilon})^{d/S})$ samples are required per subspace.
>
> Our final PQGAN subspaces have a constant dimension of 2 because of $S = d/2$ for any dimension $d$. We therefore need a constant number of observed samples to get the same latent space coverage and can utilise larger dimensions for increased representation capacity without sacrificing codebook fidelity. Meanwhile, VQ requires an exponential increase of observed training samples, which is infeasible.
>
> >*"For the mundane reader, it would be worth citing the paper that has introduced/popularized product quantization ("Product quantization for nearest neighbor search"). And typo at L329: benifits."*
>
> Thank you for the pointer, we will add the reference and fix the typo.
>
> **Questions**
> >*(1)"From Table 1, I understand that the big improvement in PSNR comes from using a resolution of 32x32, while the 16x16 patch size is only offering a PSNR of 28.3. Have you tried larger patch sizes?"*
>
> We have not evaluated larger patch sizes for our model, for the following reason. Larger patch sizes correspond to higher latent resolution and less spatial compression, which increases the computational cost of the generator. In prior settings (e.g., with weaker latent reconstructions), this trade-off was justified. However, given the fidelity of PQGAN reconstructions at $32\times32$ and even $16\times16$ latent sizes, further increasing patch size offers diminishing perceptual gains while significantly raising computational costs. We thus focus on the most efficient settings.
>
> >*(2) "Similarly, RVQ is not evaluated in the context of compression, and only reported with patch size of 8x8. So I wonder if the better performance of PQ in this context is simply due to a better hyper-parameter choice of the patch size. Have you tried replacing PQ by a RVQ?"*
>
> We do a direct comparison to RVQ. For RVQ, we take the results from the original article. The RVQ paper does not train models across different latent resolutions. To avoid introducing re-implementation bias, we instead trained PQGAN specifically for the comparison to RVQ with equivalent parameters (see Table 1, PQGAN with latent resolution $8\times8$ and L359-L361). At this resolution, PQGAN strongly outperforms RVQ for rFID and the overall metric rankings are stable across other configurations.
>
>
>
> [1] Zador, Paul. "Asymptotic quantization error of continuous signals and the quantization dimension." IEEE Transactions on Information Theory 28.2 (1982): 139-149.

---

### Meta-Review · Area_Chair_xPCk · 2026-01-05

**Summary:**

The reviewers highlighted several strengths and weaknesses.

[Strengths]
- PQGAN achieves strong empirical performance, including high reconstruction fidelity (PSNR 37.4 dB, rFID 0.036) and consistent improvements over VQ, RVQ, MCQ, and KL-AE baselines (Reviewer Nh49, gjYk, rVnn).
- The paper offers systematic and well executed analyses of scaling behavior, covering latent dimension, subspace count, and codebook size, as well as clear entropy and perplexity diagnostics (Reviewer rVnn).
- Integration into Stable Diffusion is practically valuable, enabling either doubled resolution or up to 4x faster generation at the same compute (Reviewer gjYk, Nh49, rVnn).

[Weaknesses]
- PQ is not new, and some improvements may stem from larger latent dimensionality or higher latent resolution rather than PQ itself, raising questions about novelty and fairness (Reviewer Jhoa, Nh49, rVnn).
- The theoretical explanation for why PQ scales better than VQ and exhibits opposite behavior at high dimensionality was initially limited (Reviewer rVnn).
- Generative comparisons using FID or CLIP Score for PQSD models on the same domain as SD2.1 are missing, leaving the generation quality only partially evaluated (Reviewer gjYk).

**Reviewer Concerns:**

[Addressed by rebuttal]
- rVnn’s theoretical concerns were addressed with quantization theory and sample efficiency arguments, and the reviewer expressed full satisfaction.
- Nh49’s concerns on prior PQ work will be resolved with added citations and clearer phrasing.
- gjYk’s concerns about generative metrics were partly resolved by pointing to controlled same-data FID comparisons in Appendix D.
- Concerns about codebook correlation loss and inference overhead were addressed with empirical evidence.

[Still outstanding]

I believe the remaining concerns are relatively minor and can be addressed with straightforward additions in the camera‑ready version.
- A controlled ablation matching latent dimension, latent resolution, and codebook size across PQ and VQ or KL would more cleanly isolate PQ’s intrinsic gains, but the lack of this experiment does not undermine the main empirical findings.
- The influence of latent resolution versus PQ is not fully separated, but this is a clarifiable point rather than a methodological flaw.
- Direct generative FID or CLIP comparisons on the same domain as SD2.1 are still missing, although the authors’ explanation regarding domain shift is reasonable and the controlled FID results already support the main claims.

**Reviewer Scores:**

- Reviewer rVnn: 8
- Reviewer Nh49: 6
- Reviewer gjYk: 6
- Reviewer Jhoa: 2

---

### Decision · Program_Chairs · 2026-01-26

Accept (Poster)